# Learning to Correct: Calibrated Reinforcement Learning for Multi-Attempt Chain-of-Thought

**M. Emrullah Ildiz** [* 1]   **Halil Alperen Gozeten** [* 1]   **Ege Onur Taga** [1]   **Samet Oymak** [1 2]

## Abstract

State-of-the-art reasoning models utilize long chain-of-thought (CoT) to solve increasingly complex problems using more test-time computation. In this work, we explore a long CoT setting where the model makes up to K successive attempts at solving a problem, in which each attempt is allowed to build on earlier ones after the model receives a hard verifier feedback. This motivates RL methods that can harness per-attempt rewards by carefully weighting individual attempts. We study optimizing the Verification@K reward (the model succeeds by the K-th attempt) and show that naively weighing the attempts by their pass/fail results in biased gradients. We introduce **Calibrated Attempt-Level** (CAL) GRPO by devising a weighing strategy to obtain unbiased gradients while maintaining small variance. Our theory reveals how incorporating per-attempt rewards influences the training and the eventual Verification@K performance. Experiments, baselines, and ablations on synthetic and real data corroborate our theory and the benefits of CAL-GRPO over vanilla GRPO as well as naive weighting.

**Code:** `github.com/alperengozeten/learning-to-correct`.

## 1. Introduction

The emergence of advanced reasoning models is fundamentally driven by the scaling of test-time compute, where extended CoT traces help solve more complex problems (Wei et al., 2023; Wang et al., 2023; Yao et al., 2023a; Aghajohari et al., 2025). This paradigm effectively shifts the computational burden to the autoregressive generation process, en-

abling the model to search for new approaches or revise past proposals (Yao et al., 2023b). While traditional CoT is often viewed as a single coherent trace, recent research increasingly approaches reasoning as an iterative, multi-attempt process where models utilize external verifier feedback to build on and correct earlier failures (Lightman et al., 2023; Cobbe et al., 2021; Dhuliawala et al., 2023). However, optimizing these multi-turn interactions through reinforcement learning (RL) poses a critical challenge for credit assignment (Ahmadian et al., 2024). Trajectory-level rewards fail to leverage dense feedback and can backpropagate success signals to incorrect/uninformative early attempts (Uesato et al., 2022; Lightman et al., 2023; Cobbe et al., 2021; Stiennon et al., 2022; Ouyang et al., 2022). In contrast, naive heuristics can introduce systematic biases in policy gradient estimation, motivating a principled framework (Williams, 1992; Sutton et al., 1999; Schulman et al., 2018).

In this work, we frame long CoT as a Verification@K (Ver@K) problem: The model makes at most K successive attempts until it solves the problem, and after each attempt, it gets to observe a hard pass/fail feedback. Under this setting, if a trajectory terminates at $\ell$ attempts, the model receives $\ell$ 0-1 rewards and this dense feedback can be used to enhance RL training. This motivates RL algorithms that weigh individual attempts for better policy gradient estimation. We ask:

**Q:** What RL algorithms efficiently maximize the Ver@K reward and teach the model better correction capabilities?

Specifically, we examine Attempt-Level variations of REINFORCE Leave-One-Out and GRPO where we weigh the gradient contributions at attempt-level rather than trajectory-level. An intuitive strategy is weighing the attempts by their individual pass/fail rewards, which we call Naive Attempt-Level (NAL) Algorithm. Interestingly, we establish that NAL over-emphasizes earlier attempts and results in biased policy gradients for the Ver@K objective. As our key technical contribution, we introduce C(alibrated)AL Algorithm by correcting the bias of NAL through a theoretically-grounded weight assignment.

Remarkably, we demonstrate that CAL achieves the best-of-both worlds: **(i)** It returns unbiased gradient unlike NAL and

---

[*]Equal contribution   [1]University of Michigan, Ann Arbor   [2]Google Research.   Correspondence to:  M. Emrullah Ildiz <eildiz@umich.edu>.

*Proceedings of the 43rd International Conference on Machine Learning*, Seoul, South Korea. PMLR 306, 2026. Copyright 2026 by the author(s).

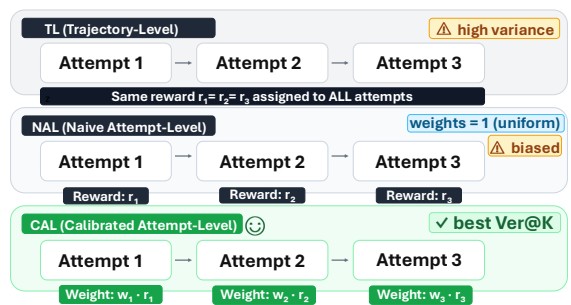

*Figure 1.* We consider RLOO and GRPO methods that weigh individual attempts. TL is the standard approach that uses the final Ver@K reward for all attempts. NAL uses per-attempt pass/fail as weights. Proposed CAL approach calibrates naive weights to obtain unbiased low-variance gradients.

**(ii)** it achieves lower variance than Trajectory-Level (TL) Algorithm, which employs trajectory-level sparse rewards. Importantly, all three variants are instantiations of a generic Attempt-Level method with specific weight assignments as depicted in Figure 1. As a remark, the number of backpropagation steps required is the same across all methods. To summarize, we make the following contributions:

i We introduce attempt-level variations of both RLOO and GRPO to facilitate finer-grained credit assignment.

ii We formally derive the policy gradient of the Ver@K objective and introduce CAL as an effective method to optimize it.

iii We establish that CAL corrects the potential bias of NAL and prove that it also achieves strictly lower variance compared to standard TL under mild conditions.

iv We demonstrate that CAL consistently outperforms TL and NAL baselines across a diverse set of reasoning tasks, including MATH, Maze navigation, and structured Markov Chain benchmarks.

## 2. Problem Setup and Preliminaries

We consider a reinforcement learning with verifiable reward (RLVR) setup where a policy model is trained to solve problems using a *verification-based multi-attempt procedure*. For each problem, the model can attempt a solution up to $K$ times, and an external hard verifier[1] provides binary feedback after each attempt. Each trajectory is assigned a single $0/1$ reward depending on whether the model achieves a correct solution within the first $K$ attempts.

Let $\mathcal{X}$ denote the space of problem inputs and $\mathcal{Y}$ the space of correct reference answers. We assume examples $(x, y)$ are drawn from a distribution $P$ over $\mathcal{X} \times \mathcal{Y}$. A policy $\pi_\theta$

---

[1]We discuss the noisy verifier case in Appendix C.

(e.g., an autoregressive language model) with parameters $\theta$ maps states to distributions over actions (token sequences).

For a fixed problem $(x, y)$, we consider the following rollout procedure. At attempt $t \in \{1, \ldots, K\}$, the environment is in state $s_t$, which encodes the original problem $x$ together with the history of previous attempts and rewards. The policy samples a candidate solution $y_t \sim \pi_\theta(\cdot \mid s_{t-1})$, and an external verifier returns a binary signal $r_t \in \{0, 1\}$ indicating whether $y_t$ is correct with respect to $y$. If $r_t = 1$, the rollout terminates immediately. If $r_t = 0$ and $t < K$, the incorrect solution $y_t$ and feedback are appended to the state, and the process continues with attempt $t + 1$. If all $K$ attempts are incorrect, the rollout terminates after $K$ attempts.

Formally speaking, let the first state $s_0$ be the problem $x$ and let $s_t = (s_{t-1}, y_t, r_t)$ be the state denoting the history up to and including the $(t)$-th attempt, and let $\tau := s_{\mathcal{T}} = (x, y_1, r_1 \cdots y_{\mathcal{T}(\tau)}, r_{\mathcal{T}(\tau)})$ denote the resulting random trajectory, where the stopping time $\mathcal{T}(\tau) \in \{1, \ldots, K\}$ is the attempt index at which the rollout stops: $\mathcal{T}(\tau)$ is the smallest index $t \le K$ with $r_t = 1$ if the model ever succeeds, and $\mathcal{T}(\tau) = K$ if the model never succeeds. We define the *Ver@K reward* for trajectory $\tau$ as $r_{\mathcal{T}(\tau)}$. For notational simplicity, we omit the trajectory $\tau$, so we use $\mathcal{T}$ instead of $\mathcal{T}(\tau)$ in this section.

For fixed $(x, y)$, let $p_\theta(\tau \mid x, y)$ denote the trajectory distribution induced by the policy $\pi_\theta$ and the verifier. For notational simplicity we write $p_\theta(\tau \mid x)$ with the dependence on $y$ through the verifier implicit. The per-example Ver@K success probability is

$$\rho_\theta(x, y) := \mathbb{E}_{\tau \sim p_\theta(\cdot \mid x)}\left[r_{\mathcal{T}}\right].$$

Our training objective is to maximize the expected Ver@K success probability:

$$J(\theta) := \mathbb{E}_{(x,y) \sim P}\left[\rho_\theta(x, y)\right]. \tag{1}$$

In this paper, we develop two different approaches to the Ver@K problem, namely trajectory-level optimization and attempt-level optimization. Next, we derive gradients for these approaches to provide insight into their behavior. In the remainder of the paper, we utilize these gradients to maximize the objective of (1) with the REINFORCE algorithm (Williams, 1992) and its extension to the leave-one-out version (Ahmadian et al., 2024).

**Trajectory-level Optimization:** In this approach, we treat each trajectory $\tau$ as a single solution sampled from the trajectory distribution $p_\theta(\cdot \mid x)$. This formulation allows us to view different trajectories as independent samples from the same distribution. Applying the log-derivative property, at the trajectory level, the gradient of (1) for a fixed problem $(x, y)$ can be written as

$$\begin{aligned}\nabla_\theta \rho_\theta(x, y) &= \nabla_\theta \mathbb{E}_{\tau \sim p_\theta(\cdot \mid x)}\left[r_{\mathcal{T}}\right] \\ &= \mathbb{E}_{\tau \sim p_\theta(\cdot \mid x)}\left[r_{\mathcal{T}} \nabla_\theta \log p_\theta(\tau \mid x)\right].\end{aligned} \tag{2}$$

The gradient is therefore an expectation over trajectories $\tau$ drawn from $p_\theta(\cdot \mid x)$ and depends on each trajectory only through its realized reward and log-probability.

**Attempt-level Optimization:** In this approach, we decompose a single trajectory into a sequence of dependent attempts and analyze each attempt individually to derive the gradient of (1) for a fixed problem instance $(x, y)$. This formulation enables attempt-level credit assignment during solution generation, improving optimization efficiency by reducing the sparsity of trajectory-level rewards. On the other hand, because the attempts are sequentially dependent, this approach requires computing more complex conditional probability distributions. The trajectory distribution (over valid trajectories) factorizes as $p_\theta(\tau \mid x) = \prod_{t=1}^{\mathcal{T}} \pi_\theta(y_t \mid s_{t-1})$, and therefore

$$\nabla_\theta \log p_\theta(\tau \mid x) = \sum_{t=1}^{\mathcal{T}} \nabla_\theta \log \pi_\theta(y_t \mid s_{t-1}). \tag{3}$$

Substituting (3) into (2) and extending the random upper limit $\mathcal{T}$ to $K$ using the indicator $\mathbf{1}_{\mathcal{T} \geq i}$ yields

$$\nabla_\theta \rho_\theta(x, y) = \sum_{i=1}^{K} \mathbb{E}_{\tau \sim p_\theta(\cdot \mid x)} \left[ \mathbf{1}_{\mathcal{T} \geq i} r_{\mathcal{T}} \nabla_\theta \log \pi_\theta(y_i \mid s_{i-1}) \right].$$

Next, define the continuation success probability

$$V_{\theta,i}(s_i) := \mathbb{E}\left[ r_{\mathcal{T}} \mid s_i \right] = \mathbb{P}(\exists t \in \{i+1, \ldots, K\} : r_t = 1 \mid s_i)$$

and $V_{\theta,K+1}(\cdot) := 0$; where $s_i$ is the state before attempt $i$. Since the verifier is deterministic, conditioning on $(s_i, y_i)$ fixes $r_i$, and the remaining Ver@K reward satisfies the one-step recursion

$$\mathbb{E}\left[ r_{\mathcal{T}} \mid s_{i-1}, y_i \right] = r_i + (1 - r_i) V_{\theta,i}(s_i),$$

where $s_i = (s_{i-1}, y_i, r_i)$ is the next state. Applying the tower property to $\nabla_\theta \rho_\theta(x, y)$ gives

$$\begin{aligned} \nabla_\theta \rho_\theta(x, y) = \sum_{i=1}^{K} \mathbb{E}_{\tau \sim p_\theta(\cdot \mid x)} \big[ &\mathbf{1}_{\mathcal{T} \geq i} \left( r_i + (1 - r_i) V_{\theta,i}(s_i) \right) \\ &\nabla_\theta \log \pi_\theta(y_i \mid s_{i-1}) \big]. \end{aligned} \tag{4}$$

In attempt-level optimization, the gradient is expressed as an expectation over attempts and depends on the continuation success probabilities $V_{\theta,i}(s_i)$.

Trajectory-level optimization requires sampling full trajectories from the underlying distribution, whereas attempt-level optimization involves computing more complex conditional probability distributions. In this paper, our goal is to develop a practical algorithm that strikes an effective trade-off between these two approaches.

In the remainder of the paper, we describe the REINFORCE algorithm with leave-one-out for both trajectory-level and

attempt-level optimization in Section 3. We then present a practical estimator for $V_{\theta,i}(s_i)$ and introduce an algorithm called *Calibrated Attempt-Level Algorithm (CAL)*. In Section 4, we compare the proposed method with existing approaches on a synthetic Markov chain benchmark, the MATH dataset, and a maze navigation task. Finally, in Section 5, we prove the unbiasedness of the proposed gradient estimator and provide variance-reduction guarantees relative to trajectory-level optimization under mild assumptions.

## 3. Derivation of RLOO Algorithms

In this section, we derive the REINFORCE with leave-one-out (RLOO) algorithm for trajectory-level and attempt-level optimization using the gradients provided in Section 2, and we study the relationship between these two approaches. Finally, inspired by these two approaches, we define a practical reward shaping function over a sampled group to apply a GRPO-based algorithm that maximizes the Verification@K reward.

We start by defining the REINFORCE with Leave-one-out (RLOO) (Ahmadian et al., 2024) algorithm for the trajectory and attempt-level optimization. While defining, we fix a problem instance $(x, y)$ and draw a *group* of $N \geq 2$ independent trajectories $\tau_1, \ldots, \tau_N \overset{\text{i.i.d.}}{\sim} p_\theta(\cdot \mid x)$.

**RLOO for Trajectory-level Optimization:** In this approach, we assign a single reward to each trajectory in order to apply the RLOO algorithm for the trajectory-level optimization. If there exists one correct answer inside the trajectory, we assign 1 reward; otherwise we assign 0 reward. Formally speaking, let $\mathcal{T}(\tau_1), \ldots, \mathcal{T}(\tau_N)$ be the corresponding stopping times for the trajectories. Then, the assigned reward for the $n$th trajectory is $r_n^{\text{TL}} := r_{\mathcal{T}(\tau_n)}$, where $r_{\mathcal{T}(\tau_n)}$ is the verification@K reward defined in Section 2. REINFORCE with Leave-One-Out (RLOO) replaces the trajectory reward $r_n^{\text{TL}}$ by the leave-one-out advantage

$$A_n^{\text{TL}} = r_n^{\text{TL}} - \frac{1}{N-1} \sum_{m \neq n} r_m^{\text{TL}},$$

and uses the trajectory-level gradient estimator

$$\widehat{G}_{\text{RLOO}}^{\text{TL}}(\theta; x, y) = \frac{1}{N} \sum_{n=1}^{N} A_n^{\text{TL}} \nabla_\theta \log p_\theta(\tau_n \mid x). \tag{5}$$

**RLOO for Attempt-level Optimization:** In this approach, we assign a reward to each attempt (instead of assigning a single reward to the whole trajectory!) using the probabilities $V_{\theta,i}(s_i)$ and $\mathbb{P}(\mathcal{T} \geq i)$. Let $r_{n,i}$ be the correctness of $n$th trajectory $i$th attempt for $n \in [N]$ and $i \in [\mathcal{T}(\tau_n)]$. Inspired by the gradient derivation for the attempt-level optimization in (4), we reshape this reward to obtain a reward for the attempt-level optimization as follows:

$$r_{n,i}^{\text{AL}} = (r_{n,i} + (1 - r_{n,i}) V_{\theta,i}(s_{n,i})) \tag{6}$$

Now, we define the advantage for each attempt $i$. The important difference from the previous approach is that we apply the leave-one-out centering for each attempt separately. Let $\mathcal{S}_i := \{n : n \in [N], \; \mathcal{T}(\tau_n) \geq i\}$. Then, when $|\mathcal{S}_i| > 1$

$$A_{n,i}^{\text{AL}} := r_{n,i}^{\text{AL}} - \frac{1}{|\mathcal{S}_i| - 1} \sum_{m \neq n, m \in \mathcal{S}_i} r_{m,i}^{\text{AL}}, \qquad (7)$$

and when $|\mathcal{S}_i| = 1$, $A_{n,i}^{\text{AL}} = 0$. Based on these advantages, we define the attempt-level gradient estimator as

$$\widehat{G}_{\text{RLOO}}^{\text{AL}}(\theta; x, y) := \sum_{i=1}^{K} \frac{1}{N} \sum_{n \in \mathcal{S}_i} A_{n,i}^{\text{AL}} \nabla_\theta \log \pi_\theta(y_{n,i} \mid s_{n,i-1}). \quad (8)$$

Until now, we have defined the RLOO algorithm for trajectory-level and attempt-level optimization. Now, we are ready to state our main result in this section:

**Theorem 3.1.** *Recall the definitions of trajectory and attempt-level estimators $\widehat{G}_{\text{RLOO}}^{TL}(\theta; x, y)$ and $\widehat{G}_{\text{RLOO}}^{AL}(\theta; x, y)$ in (5) and (8), respectively. Then, we have*

$$\mathbb{E}\left[\widehat{G}_{\text{RLOO}}^{TL}(\theta; x, y)\right] = \mathbb{E}\left[\widehat{G}_{\text{RLOO}}^{AL}(\theta; x, y)\right] = \nabla_\theta \rho_\theta(x, y).$$

*Furthermore, define $g_{n,i} := \mathbf{1}_{\mathcal{T}_n \geq i} \nabla_\theta \log \pi_\theta(y_{n,i} \mid s_{n,i-1})$ and*

$$Z_i^{\text{TL}} := \frac{1}{N} \sum_{n=1}^{N} A_n^{\text{TL}} g_{n,i}, \quad Z_i^{\text{AL}} := \frac{1}{N} \sum_{n=1}^{N} A_{n,i}^{\text{AL}} g_{n,i}.$$

*Then, we have*

$$\text{Cov}(Z_i^{\text{TL}}) \succeq \text{Cov}(Z_i^{\text{AL}}), \qquad \forall i \in [K-1],$$

*where the equality occurs only when $K = 1$ or $V_{\theta,i}(s_i) \in \{0, 1\}$ for all possible $s_i$. Equivalently, for every direction $v$,*

$$\text{Var}(\langle v, Z_i^{\text{TL}} \rangle) \geq \text{Var}(\langle v, Z_i^{\text{AL}} \rangle).$$

This theorem yields two conclusions. First, the RLOO gradient estimators for trajectory-level and attempt-level optimization are unbiased. The key reason is that we define the rewards $r_i^{\text{TL}}$ and $r_{n,i}^{\text{AL}}$ to match the gradients derived in (2) and (4). Second, the attempt-level gradient estimator has lower variance than the trajectory-level gradient estimator, because it exploits the rewards from individual attempts rather than treating the entire trajectory as a single sample. Under reasonable assumptions on the attempt-level components, we show the variance reduction for each attempt. Under suitable cross-covariance conditions between the attempts' gradients, it implies that $\text{Cov}(\widehat{G}_{\text{RLOO}}^{\text{TL}}) \succeq \text{Cov}(\widehat{G}_{\text{RLOO}}^{\text{AL}})$. The proof for the above theorem is provided in Appendix C.

The main practical drawback of attempt-level optimization is that we assume access to the probabilities $V_{\theta,i}(s_{n,i})$ for every possible trajectory. In practice, it is difficult to compute these probabilities for all trajectories. To overcome this

difficulty, we provide a rough estimate of these probabilities. We refer to this algorithm as the Calibrated Attempt-level algorithm.

**Calibrated Attempt-level Algorithm (CAL):** In this algorithm, we fix a problem $(x, y)$ and draw a *group* of $N$ independent trajectories $\tau_1, \ldots, \tau_N \overset{\text{i.i.d.}}{\sim} p_\theta(\cdot \mid x)$. Recall that $r_{n,i}$ represents the correctness of the output for the $n$th trajectory at $i$th attempt and recall the definition of $\mathcal{S}_i = \{n : n \in [N], \; \mathcal{T}(\tau_n) \geq i\}$. We define an estimate of the success probability for the attempt $i \in [\mathcal{T}(\tau_n)]$ with leave-one-out as

$$\hat{\rho}_{n,i} := \frac{\sum_{m \in \mathcal{S}_i, m \neq n} \mathbf{1}_{r_{m,i}=1}}{|\mathcal{S}_i| - 1}.$$

Then, inspired by the reward definition $r_{\theta,i}^{\text{AL}}$ in (6), we assign the following weight to the attempt $i \in [K]$:

$$w_{n,i}^{\text{CAL}} = \Pi_{j=i+1}^{K}(1 - \hat{\rho}_{n,j}), \qquad (9)$$

For the reward $r_{n,i}^{\text{CAL}} := r_{n,i}$ and define $A_{n,i}^{\text{CAL}}$ accordingly similar to (7). Then, the gradient estimator for the weighted attempt-level algorithm is the following:

$$\widehat{G}_{\text{RLOO}}^{\text{CAL}}(\theta; x, y)$$
$$:= \sum_{i=1}^{K} \frac{1}{N} \sum_{n \in \mathcal{S}_i} w_{n,i}^{\text{CAL}} A_{n,i}^{\text{CAL}} \nabla_\theta \log \pi_\theta(y_{n,i} \mid s_{n,i-1}). \quad (10)$$

Note that we do not utilize the probability of $V_{\theta,i}(s_i)$ in the definition of (10) compared to (6) as we estimate the future success probabilities based on the samples inside the sampled. In addition to this algorithm, we also defined two different baselines.

**Naive Attempt-level Algorithm (NAL):** In this algorithm, we assign a reward to each attempt in a trajectory, but we do not apply the weight that we defined in (9). Instead, we weight with 1 and obtain the gradient estimator for the attempt-level algorithm similar to (10).

**Trajectory-level Algorithm (TL):** In this algorithm, we directly use the gradient estimator defined in (5).

An important remark is that the number of backpropagation steps required is the same across all methods. Compared to the TL algorithm, the NAL and CAL algorithms assign rewards to individual attempts rather than a single reward to the entire trajectory, which reduces the variance of the gradient estimator (Theorem 3.1). The key distinction between the two is that the CAL algorithm assigns attempt-specific weights based on future success probabilities, inspired by (6), whereas the NAL algorithm uses uniform weights. A central contribution of this paper is this calibrated attempt-wise reward assignment, which avoids treating each trajectory as a single output generation. In our experiments,

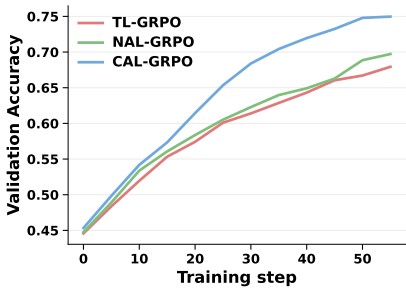 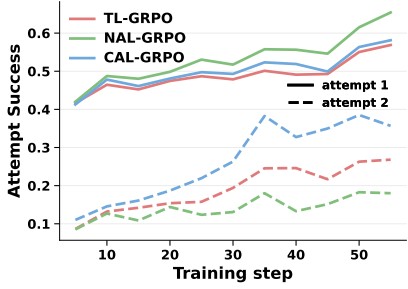 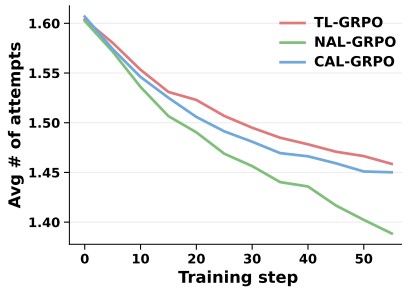

*(a)* **Ver@2 validation accuracy on MATH** across training. CAL-GRPO attains the highest Ver@2, with NAL-GRPO improving over TL-GRPO.

*(b)* **Attempt-wise success.** Solid: attempt-1 success. Dashed: attempt-2 success conditioned on attempt 1 failing; CAL-GRPO notably improves corrections on attempt-2.

*(c)* **Average attempts per problem** under Ver@2. NAL-GRPO reduces attempts the most, while CAL-GRPO trades slightly higher attempts for higher Ver@2.

*Figure 2.* **Optimizing the Ver@2 objective on MATH with GRPO variants.** We compare Trajectory-Level GRPO (TL-GRPO), Naive Attempt-Level GRPO (NAL-GRPO), and Calibrated Attempt-Level GRPO (CAL-GRPO). **(a)** CAL-GRPO achieves the best Ver@2 over training. **(b)** The improvement is primarily driven by stronger second-attempt correction, whereas NAL-GRPO emphasizes first-attempt success. **(c)** NAL-GRPO minimizes the number of attempts, while CAL-GRPO uses slightly more attempts but delivers higher overall Ver@2.

the NAL algorithm serves as a natural ablation, representing a naive uniform-weight variant of the proposed CAL algorithm.

# 4. Experiments on Ver@K

In this section, we evaluate TL-GRPO, NAL-GRPO, and CAL-GRPO on three benchmarks: a synthetic Markov chain task, MATH, and Maze. We first describe the experimental setup and then present the main results and analysis on Ver@K.

## 4.1. Experimental Setup

We compare three algorithms, namely trajectory-level, naive attempt-level, and calibrated attempt-level on three datasets: a synthetic Markov chain task, the MATH dataset (Hendrycks et al., 2021), and the Maze navigation task (Ivanitskiy et al., 2023). All algorithms are implemented using Group Relative Policy Optimization (GRPO) (Shao et al., 2024) with the advantage shaping methods described in Section 3. Different from the previous section's leave-one-out formulation, in our experiments we use the standard GRPO group normalization: advantages are computed by centering rewards with the within-group empirical mean and scaling by the within-group empirical standard deviation; for NAL-GRPO and CAL-GRPO this normalization is applied separately at each attempt index over trajectories that reach that attempt. Throughout the main experiments, we use this GRPO normalization because scaling by the within-group empirical standard deviation improves sample efficiency in practice. Appendix B ablates this choice in Figure 15 and separately studies rollout group size in Figure 14, which is particularly relevant for CAL-GRPO because its calibrated weights depend on empirical future-

attempt success estimates computed from the current rollout group, and these estimates become more accurate as the group size increases. Following the notation in Section 3, in this section we refer to these algorithms as TL-GRPO, NAL-GRPO, and CAL-GRPO, respectively. Further benchmark, verifier, rollout, and GRPO-variant details are provided in Sections A.2 to A.5.

For the MATH and Maze tasks, we implement Ver@K training by extending the open-source verl framework (Sheng et al., 2025) and leveraging its multi-attempt rollout and interaction framework for iterative verifier feedback. Specifically, we use a multi-attempt interaction loop (problem → attempt → verifier feedback) with $K \in 2, 4$ (Ver@2 and Ver@4), while keeping rollout and group-sampling configurations fixed across all estimators. For the Maze Ver@2 results on 5×5 and 9×9, we report mean curves over five independent training runs with different random seeds, together with confidence intervals across runs. For the remaining settings, we report single-run curves.

**MATH Dataset.** We train and evaluate on the MATH dataset (Hendrycks et al., 2021), which contains 12,500 competition-style problems (7,500 train / 5,000 test). We use the official train split for RL training and report Ver@K on the held-out test split. The verifier extracts the model's final answer and compares it to the ground-truth answer (reward $r_t=1$ if correct, else 0). When an attempt is incorrect, we append a short, fixed-format feedback message indicating failure and prompt the model to revise its answer and try again; otherwise the interaction terminates immediately on success.

**Maze Task.** We consider a text-based maze navigation task which is a standard benchmark of planning and sequential decision-making for LLMs (Dao & Vu, 2025; Chen et al.,

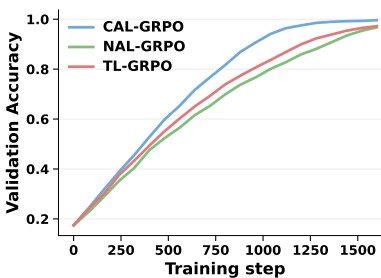 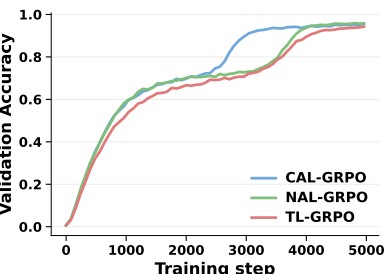 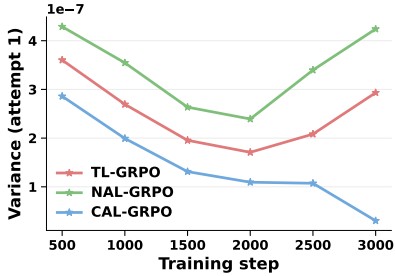

*(a)* **No-trap Markov chain Ver@4** ($n_{\text{hubs}}$=5, $m$=6). Validation success vs. training step; the calibrated attempt-level method learns fastest and reaches the highest success.

*(b)* **Trap Markov chain Ver@2** ($n_{\text{hubs}}$=5, $m$=6). Validation success vs. training step; the calibrated attempt-level method reaches higher accuracy earlier.

*(c)* **Trap Markov chain Ver@2** ($n_{\text{hubs}}$=5, $m$=6). Variance of the first attempt vs. training step; the calibrated attempt-level method has the lowest variance.

*Figure 3.* **Optimizing the Ver@4 and Ver@2 objectives on Markov Chain task with GRPO variants.** We compare TL-GRPO, NAL-GRPO, and CAL-GRPO. Success is defined by reaching the absorbing terminal state with a valid shortest-length trajectory; the trap setting in (b) additionally fails any attempt that visits the trap state. CAL-GRPO consistently reaches higher accuracies in both settings and breaks the plateau earlier than TL-GRPO and NAL-GRPO in the trap setting.

2025; Ivanitskiy et al., 2023). We generate synthetic datasets using the configurable `maze-dataset` library (Ivanitskiy et al., 2023). For maze sizes 9×9, 7×7, and 5×5, we create 10,000 training and 1,000 test mazes. Each instance provides a maze grid with a designated start and goal. The model outputs a sequence of discrete moves of {U,D,L,R}; and we deterministically check whether the sequence constitutes a valid path from start to goal without crossing walls, and return $r_t \in \{0, 1\}$. We train with the same Ver@K procedure and the same three GRPO estimators as in the previous sections, and report Ver@K on the corresponding test sets for each maze size. We provide further experimental details in Appendix A.

**Markov Chain Task.** We introduce a synthetic structured Markov chain task for controlled evaluation of multi-attempt reward assignment under a deterministic verifier. The state space is organized into $n_{\text{hubs}}$ ordered hubs, each containing $m$ states, so that $n_{\text{states}} = n_{\text{hubs}} \cdot m$. Transitions are predominantly local within a hub (moves to nearby states, e.g., within ±1 positions), with additional sparse *boundary* transitions that connect consecutive hubs. The final state in the last hub is an absorbing terminal state. Each instance samples a start state $s_0$ *from the set of states that can reach the terminal*. In each attempt, the model autoregressively proposes a candidate state sequence. The verifier returns $r_t = 1$ iff the proposed trajectory (i) reaches the absorbing terminal state, (ii) uses only valid transitions in the Markov chain transition graph, and (iii) has length equal to the precomputed shortest-path distance from $s_0$ to the terminal. We also consider a *trap* variant, where a trap state is sampled among two pre-terminal states in the last hub with 1/2 probability. Visiting the trap causes immediate failure, and shortest-path distances are computed on the graph with the trap removed. Under Ver@K, failed attempts reset to the same start state,

and the model retries for up to $K$ attempts with early stopping. The trapped version needs Ver@K ($K \geq 2$) approach to solve the problem as the trap is randomly selected for each trajectory.

**Models.** For MATH and Maze, we train the instruction-tuned Qwen3-4B-Instruct model (Yang et al., 2025). For the Markov chain benchmark, we initialize the policy from an SFT checkpoint trained on synthetic optimal trajectories generated from the known transition graph. Concretely, we sample start states (and also the trap state in the trap setting), compute the corresponding shortest valid trajectory to the absorbing terminal state on the graph with the trap removed when applicable, and use the resulting state-id sequence as the supervised target. We then fine-tune this checkpoint with the same Ver@K GRPO procedure used in the main experiments. Sections A.2 to A.4 provide the benchmark, verifier, tokenization, rollout, and optimization details.

### 4.2. Results and Analysis

We study three questions: (Q1) Does *attempt-level credit assignment* (NAL-GRPO/CAL-GRPO) improve over *trajectory-level credit assignment* (TL-GRPO)? (Q2) Is the calibration in Eq. (10) necessary beyond naive attempt-level rewards? (Q3) How does calibration allocate learning between solving earlier and correcting later, and what trade-off does this induce between final Ver@K accuracy and expected test-time attempts under early stopping? On MATH and Maze, attempt-level credit assignment substantially improves over TL-GRPO. On the Markov chain task, however, naive attempt-level normalization is not uniformly beneficial and can underperform TL-GRPO. Across all three benchmarks, CAL-GRPO is the most robust method because Eq. (10) reweights each attempt by its estimated marginal contribution to final Ver@K.

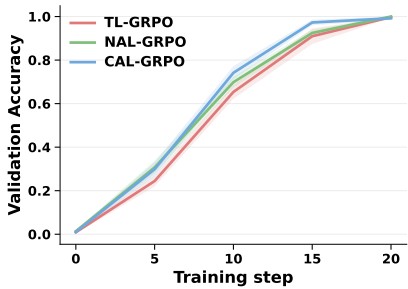

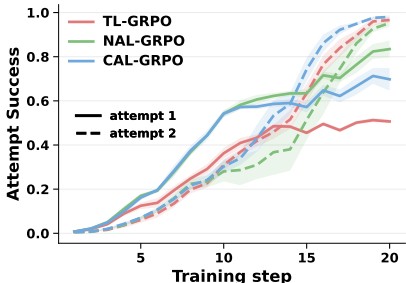

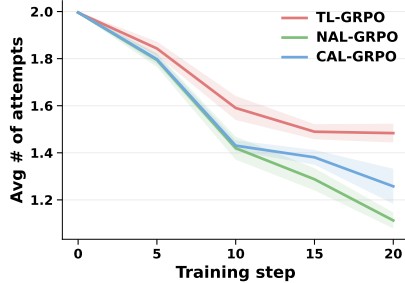

*(a)* **Ver@2 validation accuracy on Maze** across training; CAL-GRPO reaches higher accuracy earlier, and all methods converge near-perfect.

*(b)* **Attempt-wise success.** Solid: attempt-1 success. Dashed: attempt-2 success given attempt-1 failure; CAL-GRPO yields the strongest correction on the second attempt.

*(c)* **Average attempts per instance** under early stopping. NAL-GRPO minimizes the number of attempts, mimicking optimization of a discounted objective.

*Figure 4.* **Optimizing the Ver@2 objective on** 5×5 **Maze with GRPO variants.** We compare Trajectory-Level GRPO (TL-GRPO), Naive Attempt-Level GRPO (NAL-GRPO), and Calibrated Attempt-Level GRPO (CAL-GRPO). **(a)** CAL-GRPO reaches higher Ver@2 accuracy earlier in training. **(b)** NAL-GRPO increases attempt-1 success, while CAL-GRPO strengthens attempt-2 correction. **(c)** NAL-GRPO uses the fewest attempts, effectively behaving like optimization of a discounted objective.

### 4.2.1. MATH AND MAZE TASKS

**Attempt-level credit assignment improves learning over trajectory-level credit assignment.** Figures 2 and 4 show that, on MATH and Maze, moving from trajectory-level credit assignment (TL-GRPO) to attempt-level credit assignment (NAL-GRPO/CAL-GRPO) substantially improves learning under Ver@2 and Ver@4 objectives (please see Appendix B for Ver@4 results). TL-GRPO assigns the same binary trajectory success signal to all tokens, so a trajectory solved only on attempt 2 still provides positive updates to the attempt-1 tokens that produced an incorrect first answer. Under multi-attempt procedure, this mixes gradients for first-attempt correctness and correction-with-feedback, yielding a noisy (and sometimes conflicting) learning signal for improving attempt-1 behavior. In contrast, NAL-GRPO normalizes rewards separately per attempt among trajectories that actually reach that attempt, which yields cleaner reward assignment: attempt 1 is trained on first-attempt correctness, and attempt 2 is trained on correction conditioned on failure. This sharply accelerates attempt-1 success and correspondingly reduces the expected number of attempts executed under early stopping (Figures 2c and 4c).

**Naive attempt-level normalization induces an early-attempt bias.** However, NAL-GRPO implicitly down-weights later attempts as training progresses: because attempt $t+1$ is visited only on the shrinking subset of failure cases, the total gradient contribution of later attempts is effectively multiplied by a reach probability, which prioritizes earlier-attempt correctness to minimize attempts, closely aligning with the next section of Corollary 5.3. This indicates that NAL-GRPO implicitly optimizes a discounted objective where later steps receive smaller weight once earlier steps become more accurate, where the discount factors depend on the success rate of the attempts.

**Calibration restores learning signal for later-attempts and improves final Ver@K.** Our *Calibrated Attempt-Level GRPO (CAL-GRPO)* corrects this misalignment by reweighting attempts according to Eq. (9), which estimates the probability that all later attempts fail and therefore measures the marginal contribution of attempt $t$ to final Ver@K success. Intuitively, this focuses attempt-1 updates on the hard residual cases where a second attempt is unlikely to correct the mistake anyway, while avoiding over-investing in attempt-1 improvements when attempt 2 already succeeds accurately, where further gains in attempt-1 accuracy have diminishing returns for Ver@2 accuracy. As a result, CAL-GRPO preserves more update budget for learning correction-with-feedback when second-attempt correction still contributes substantially to Ver@2. Empirically, CAL-GRPO yields the best Ver@2 on both tasks (Figures 2a and 4a) by substantially increasing conditional second-attempt success (Figures 2b and 4b), at the cost of a modest increase in average attempts relative to NAL-GRPO (Figures 2c and 4c). This indicates that if the primary goal is Ver@K accuracy, the attempt-weighted CAL-GRPO method works well; and if reducing number of attempts matters, NAL-GRPO can be preferable. We provide further analysis and additional Maze results for 9×9 and 7×7 under Ver@2 and Ver@4 in Appendix B.

### 4.2.2. MARKOV CHAIN TASK

**A different regime under stricter verifier.** Figure 3 provides a controlled comparison of TL-GRPO, NAL-GRPO, and CAL-GRPO and highlights a qualitatively different regime from MATH/Maze. We observe that NAL-GRPO is the least sample-efficient and underperforms even TL-GRPO in both the no-trap and trap settings. With a strict, sequence-level verifier (valid shortest path or nothing), per-attempt normalization can over-penalize early failed at-

tempts and slow down learning progress. Under NAL-GRPO, attempt-1 tokens receive zero/negative advantage whenever attempt 1 fails, *even if the trajectory is "useful" in the sense that it enables a later attempt to succeed after feedback*. On the other hand, TL-GRPO still propagates eventual Ver@K success back to earlier tokens, which can implicitly reward early-attempt behaviors that set up effective correction, and can therefore remain competitive.

**Calibration preserves useful correction signal.** CAL-GRPO consistently outperforms the other methods as it keeps attempt-aware reward assignment while explicitly reweighting updates toward the later attempts that still contribute the most to marginal Ver@K success by effectively preserving learning budget for the correction behavior that drives the remaining improvements.

**CAL-GRPO breaks the trap-induced plateau earlier.** In the *trap* variant of the Markov chain task, training exhibits a plateau around $\approx 0.75 = 1 - (1/2)^2$ as trap state is random with 1/2 probability before further gains, which reflects the additional late-stage failure mode introduced near the terminal hub. The policy can learn the general "forward shortest-path" behavior relatively quickly, but accurately avoiding the trap while still satisfying the shortest-path constraint is harder and becomes the dominant error source. CAL-GRPO breaks through this plateau earlier than the other baselines, which matches its intended behavior. Once attempt-1 behavior is already reasonably good, further improvements in Ver@K come from stabilizing correction behavior on retries for the remaining hard cases (here, trap-related failures), rather than continuing to over-optimize the first attempt.

**CAL-GRPO lowers gradient-estimator variance.** Figure 3c reports the variance of the first-attempt gradient estimate, and Figure 13 provides the corresponding full-gradient bias and variance diagnostics on the trap Markov-chain task. We share the details of how we calculate the variance and bias of the gradient estimators in Appendix B. The empirical pattern matches the theory: (i) TL-GRPO is closest to the true Ver@K gradient but has high variance because it assigns the same terminal success bit to all attempts. (ii) CAL-GRPO keeps the variance benefit of attempt-level credit assignment while correcting much of this bias through future-failure reweighting, which is consistent with Theorem 3.1. (iii) NAL-GRPO reduces reward sparsity but introduces the early-attempt bias, which will be discussed in the next section, specifically predicted by Corollary 5.3. Consequently, Figure 3c illustrates the variance ordering as CAL-GRPO < TL-GRPO < CAL-GRPO. Figure 13a shows the bias ordering TL-GRPO < CAL-GRPO < NAL-GRPO, while Figure 13b shows that CAL-GRPO has the lowest overall variance. The late rise in the second-attempt variance of CAL-GRPO in Figure 13d occurs after the trap-induced plateau is broken, when second attempts become

strongly coupled to the first-attempt failure mode. This is precisely the regime where the attempt-index independence assumption in Assumption 5.1 becomes less accurate, which will be discussed in the next section.

# 5. Optimization of Verification@K at Attempt Level

In Section 3, we have compared the trajectory-level optimization and attempt-level optimization, and we have provided a practical estimate of the attempt-level optimization with the weighted attempt-level algorithm. In this section, we prove the unbiasedness of the weighted attempt-level algorithm and establish variance reduction guarantees under the independent attempt index assumption we state below. Furthermore, we provide more insights about the attempt-level algorithm that we described in Section 3.

The independent attempt-index assumption is the following:

**Assumption 5.1.** For a fixed problem $(x, y)$ and any attempt index $i \in \{1, \ldots, K\}$, we assume that the conditional distribution of the $i$-th response depends only on $x$ and the number of previous responses. Formally, for any two histories $s_{1:i-1}$ and $s'_{1:i-1}$ of the same length,

$$p_\theta(y_i = y \mid s_{1:i-1}) = p_\theta(y_i = y \mid s'_{1:i-1}) \quad \text{for all } y.$$

Assumption 5.1 implies that there exists a family of policies $(\pi_\theta^{(i)}(\cdot \mid x))_{i=1}^K$ such that

$$p_\theta(y_i = y \mid s_{1:i-1}) = \pi_\theta^{(i)}(y \mid x), \quad i = 1, \ldots, K.$$

Now, we are ready to state the theorem:

**Theorem 5.2.** *Suppose that Assumption 5.1 holds. Recall the definition of $\widehat{G}_{\text{RLOO}}^{\text{CAL}}(\theta; x, y)$ in (10). Then, we have the following:*

$$\mathbb{E}[\widehat{G}_{\text{RLOO}}^{\text{CAL}}(\theta; x, y)] = \nabla_\theta \rho_\theta(x, y).$$

This theorem is a conjugate of Theorem 3.1 for the calibrated attempt-level algorithm under Assumption 5.1. It shows that the calibrated attempt-level algorithm is unbiased. The proof is provided in Appendix D. Next, we analyze the naive attempt index model. Before, define its single-attempt success probability for each attempt $i$,

$$\rho_{\theta,i}(x, y) := \mathbb{P}\left(r_i = 1 \mid x, y, \text{using policy } \pi_\theta^{(i)}\right).$$

As before, $r_i \in \{0, 1\}$ indicates whether the $i$-th response is correct with respect to the answer $y$.

**Corollary 5.3.** *The naive attempt-level algorithm optimizes $\sum_{i \in [K]} \gamma_i \mathbb{P}(r_i = 1)$ in expectation where $\frac{\gamma_i}{\gamma_j} = \Pi_{t=j+1}^i (1 - \rho_{\theta,i}(x, y))$ for $i > j$.*

This corollary is a direct result of the unbiasedness of $\widehat{G}_{\text{RLOO}}^{\text{CAL}}(\theta; x, y)$ by Theorem 5.2 and the definition of $w_{n,i}^{\text{CAL}}$ in (9). It shows that the naive attempt-level algorithm optimizes a discounted loss as a function of $K$ instead of the Verification@K loss defined in (1), which is consistent with Figures 2c and 4c. It also shows that the naive attempt-level algorithm is biased whereas the proposed calibrated attempt-level algorithm is shown to be unbiased in Theorem 5.2.

## 6. Related Work

**Reinforcement learning with verifiable feedback for reasoning LLMs.** Policy-gradient fine-tuning of reasoning LLMs with automatically verifiable rewards has been actively explored across a range of settings (Guo et al., 2025; Cui et al., 2025; Heckel et al., 2026; Wang et al., 2025; Zhang et al., 2025b; Su et al., 2025; Gozeten et al., 2026). Several analyses further study what such training changes in the model's solution distribution and how it interacts with pass@k-style evaluation (Yue et al., 2025; Wen et al., 2025). Complementary work improves optimization stability via refined reward assignment or denser intermediate learning signals (Kazemnejad et al., 2025). In contrast, our work explicitly targets sequential retries with dependent attempts and directly optimizes a Verification@K objective with attempt-level weighting to better match the long-CoT setting.

**Multi-attempt and self-feedback for revision.** Multi-attempt prompting and retry-based mechanisms can elicit iterative reasoning and answer revision in LLMs (Liu et al., 2025; Zelikman et al., 2022). Self-feedback and tool-assisted critique loops further enable iterative refinement and reflection without requiring per-instance human corrections (Madaan et al., 2023; Gou et al., 2024; Shinn et al., 2023). At the same time, several works characterize when self-correction succeeds or fails, emphasizing the importance of reliable external feedback (Huang et al., 2024).

**Multi-turn RL and fine-grained reward assignment.** A line of work studies RL for long-horizon, multi-turn LLM agents, where sparse terminal rewards make reward assignment difficult. In search and tool-use settings, recent methods add turn-level verifiable or judge-based rewards, intrinsic information-gain rewards, or finer step-level advantages built from grouped trajectories, shared states, or trajectory graphs (Wei et al., 2025; Wang et al., 2026; Feng et al., 2025; Li et al., 2025). In code and tool-integrated reasoning, related approaches leverage execution feedback, learned verifier scores, single-step rewards, discounted turn returns, reward shaping, or feedback-conditioned rollout trees to support iterative self-correction (Jain et al., 2025; Ding et al., 2025; Ekbote et al., 2026). Unlike these works, which mostly improve generic long-horizon agent training with task-specific turn- or step-level rewards, we study repeated attempts on the same input under hard verifier feedback and derive attempt-level weighting specifically for optimizing Verification@K.

**Verification@K, best-of-$N$, and verifier-guided test-time scaling.** Verifier-guided selection and reranking, such as Best-of-$N$ and Pass@$K$, along with training objectives tailored to optimize these criteria, form a common approach for improving reasoning reliability (Cobbe et al., 2021; Zhang et al., 2025a; Thrampoulidis et al., 2025; Mahdavi et al., 2025). Process-level supervision and step-wise verifiers enable richer guidance signals for search and reasoning-time allocation (Lightman et al., 2023; Khalifa et al., 2025; Park et al., 2024). More broadly, work on test-time compute studies how to allocate sampling, search, and deliberation budgets and how to adapt models at test time for improved accuracy–compute tradeoffs (Snell et al., 2024; Wang et al., 2023; Yao et al., 2023a; Gozeten et al., 2025).

## 7. Conclusion

In this paper, we introduced a formal view of long chain-of-thought as a *multi-attempt* process with *dependent* samples under verifier feedback, and studied direct optimization of the resulting Verification@K objective. From the corresponding policy-gradient structure, we derived attempt-aware credit assignment and proposed *Calibrated Attempt-Level algorithm* (CAL), which reweights per-attempt learning signals to better match each attempt's marginal contribution to Ver@K while keeping updates well-scaled.

Across a controlled Markov chain benchmark (including a trap variant) and real reasoning tasks (MATH and text-based Maze navigation), our empirical results show a consistent pattern: trajectory-level training (TL) entangles first-attempt correctness with later correction behavior, while naive attempt-level training (NAL) yields cleaner optimization and faster improvements in early-attempt success and efficiency in number of attempts. CAL further improves final Ver@K by sustaining useful learning pressure on later attempts when they remain the primary driver of additional gains, revealing an accuracy–compute trade-off that can be tuned depending on whether peak Verification@K or fewer retries is the priority.

A natural next step is to extend calibrated attempt-level weighting beyond hard binary verifiers to noisy/learned verifiers[2], and to characterize when unbiasedness and variance-reduction guarantees still hold. It would also be valuable to scale to larger $K$ and study adaptive compute allocation (adaptive $K$ values) rather than using a fixed number of attempts.

---

[2]Please find Appendix C for an initial effort to the noisy verifier case

## Acknowledgments

This work is supported in part by the NSF grants CCF-2046816, CCF-2403075, and CCF-2212426, and by the ONR grant N00014-24-1-2289.

## Impact Statement

This work studies reward assignment for multi-attempt reasoning with hard verifier feedback. A potential positive impact is improved reliability in domains where answers can be externally verified, since the method better aligns training with success across repeated attempts. Potential risks are that stronger multi-attempt reasoning systems can also strengthen general-purpose automation and misuse, and that repeated attempts increase training and inference compute. Our current analysis and experiments focus on deterministic binary verifiers, deployment with noisy or learned verifiers should therefore be evaluated carefully because the theoretical guarantees in this paper do not directly cover that setting. All experiments use public or synthetic data and do not involve human subjects.

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

# A. Experimental Details

## A.1. Implementation of Ver@K GRPO Training

We train policies with Ver@K using a multi-turn interaction loop implemented on top of the veRL/HybridFlow framework (Sheng et al., 2025; Volcengine, 2024) and its multi-turn interaction system with an SGLang rollout backend (Zheng et al., 2024). We modify the veRL source code to support (i) multi-attempt rollout with the same maximum output length and (ii) attempt-wise GRPO reward normalization and reweighting for the estimators described in Sections 3 and 4.

Each rollout allows up to $K$ assistant attempts. After attempt $k$, a deterministic verifier returns a binary reward signal $r_k \in \{0, 1\}$. If $r_k = 1$, the interaction terminates. Otherwise, if attempts remain, we inject a fixed feedback message indicating the attempt was incorrect and request another attempt. Attempts not reached due to early stopping are excluded from attempt-wise normalization at that step.

## A.2. Datasets and Prompting

**MATH.** We train and evaluate on the MATH dataset (Hendrycks et al., 2021). Each example is converted into a single-attempt chat prompt consisting of the problem statement followed by the instruction: *"Let's think step by step and put the final answer in \boxed{}."*

**Maze.** We generate maze navigation datasets using the `maze-dataset` library (Ivanitskiy et al., 2023). For each maze size (5×5, 7×7, 9×9), we generate 10k training and 1k test instances. Mazes are sampled with the library's randomized depth-first-search lattice generator and converted to a text representation for prompting (Ivanitskiy et al., 2023). We follow prior work studying maze navigation and multi-sample objectives for reasoning models (Dao & Vu, 2025; Chen et al., 2025). Each maze prompt provides an ASCII grid with walls/free cells and explicit start/goal markers. The model is instructed to output a move sequence inside `<answer>...</answer>` using the alphabet {U,D,L,R}.

**Markov chain.** We construct a synthetic planning benchmark based on a structured Markov chain, designed to isolate multi-attempt credit assignment under a deterministic verifier with controllable difficulty. The state space is organized into $n_{\text{hubs}}$ ordered hubs with $m$ states per hub ($n_{\text{states}} = n_{\text{hubs}} \cdot m$). The transition graph is fixed across instances: transitions are predominantly local within a hub (e.g., to nearby states within ±1 positions), with sparse *boundary* transitions connecting consecutive hubs; the final state in the last hub is an absorbing terminal state. Each instance samples a start state $s_0$ from the set of states that can reach the terminal. In each attempt, the model generates a token sequence representing a candidate state trajectory, and the verifier simulates it on the transition graph and returns $r_t = 1$ iff the trajectory reaches the terminal, uses only valid transitions, never decreases the state id, and has length equal to the precomputed shortest-path distance from $s_0$ to the terminal (i.e., it is an optimal-length path). We train with Ver@K by separating attempts with a special RETRY token; after a failed attempt, the interaction resets to the same start state and the model retries up to $K$ attempts with early stopping. We also consider a *trap* variant where a trap state is sampled per rollout from two pre-terminal states in the last hub; visiting the trap causes immediate failure, and shortest-path distances are computed on the transition graph with the trap removed.

## A.3. Verifiers and Rewards

**MATH verifier.** We extract the final boxed expression from the model output, apply normalization, and check equivalence against the ground-truth answer. The per-attempt reward is $r_k = 1$ if correct and 0 otherwise.

**Maze verifier.** We extract the move string from `<answer>` tags, normalize to {U,D,L,R}, and simulate the trajectory on the maze. The per-attempt reward is $r_k = 1$ iff the move sequence reaches the goal without crossing walls or leaving the grid; otherwise 0.

**Markov chain verifier.** Given a proposed state sequence for an attempt, we simulate it on the chain and return $r_k \in \{0, 1\}$. An attempt receives $r_k = 1$ iff it reaches the absorbing terminal state, all transitions are valid under the chain's transition structure, the state id never decreases, and the attempt length equals the precomputed shortest-path distance from the start state. In the trap variant, visiting the sampled trap state causes immediate failure and shortest-path distances are computed with the trap removed.

## A.4. Rollout, Sampling, and Optimization

**Model and algorithm.** For MATH and Maze, we fine-tune Qwen3-4B-Instruct-2507 (Yang et al., 2025) using GRPO (Shao et al., 2024), implemented in veRL (Sheng et al., 2025; Volcengine, 2024) with our multi-turn and estimator modifications (Appendix A.5). For the Markov chain task, we initialize the policy from an SFT checkpoint and then apply the same Ver@K GRPO procedure.

**Markov-chain SFT initialization.** The SFT checkpoint used to initialize the Markov-chain runs is trained on synthetic Markov chain trajectories. For each synthetic instance, we sample a valid Markov chain trajectory from the ground-truth Markov chain with a random length. In these Markov chain trajectories, there is no condition that the end state is the terminal state and that the trajectory is the shortest path to the end state. The aim of these synthetic trajectories is to train the model that knows the Markov chain transitions. This SFT checkpoint is used only as initialization for the trap and no-trap versions, and all TL-GRPO, NAL-GRPO, and CAL-GRPO runs then use the same Ver@K GRPO setup described below. Using GRPO, we train the model to follow the tasks. For the no-trap setting, the task is to follow the shortest path to the terminal state from a given start state. For the trap setting, the task is to follow the shortest path to the terminal state during which it does not visit the trap state. There is a growing literature of teaching LLM to follow Markov chain transitions (Ildiz et al., 2024).

**Rollout and decoding.** We use SGLang for rollout generation (Zheng et al., 2024). Decoding uses temperature 0.6, top-$p$ 0.95, and top-$k$ 20. The GRPO group size (number of rollouts per prompt) is $N$=20 for MATH and $N$=16 for Maze. We evaluate every 5 iterations. Unless otherwise stated, reported curves correspond to a single training run. For the 5×5 and 9×9 Maze Ver@2 experiments, we repeat training with five random seeds and report the mean together with 95% confidence intervals across runs. We focus this multi-seed evaluation on these Maze Ver@2 settings due to computational constraints.

For Markov chain setting, we generate rollouts with a lightweight token-level simulator (no tool calls): the model emits discrete state-id tokens in $\{0, \ldots, n_{\text{states}}-1\}$, with attempts separated by an injected RETRY token that resets the attempt to the same start state. Concretely, the RETRY token id is set to $n_{\text{states}}$ and the PAD token id to $n_{\text{states}}+1$, where $n_{\text{states}} = n_{\text{hubs}} \cdot m$. We sample with temperature 1.0 and disable top-$k$ truncation (top-$k$=0). For each update, we sample a batch of start states, repeat each start state to form a rollout group, and roll out $K$ attempts per start. In the trap variant, start states are sampled only from hubs before the last hub so that the sampled trap remains ahead in the chain.

**Sequence lengths.** We cap each attempt to 512 output tokens for MATH and for the 5×5 Maze task, and to 1024 new tokens for the 9×9 Maze task. With $K$=2, the total assistant budget is 1024 tokens (MATH and 5×5 Maze) or 2048 tokens (9×9 Maze) plus a small buffer for system/feedback, yielding a maximum rollout length of 1280 tokens (MATH and 5×5 Maze) or 2304 tokens (9×9 Maze).

For the Markov chain setting, we cap the length of each attempt to a fixed token budget. By default this budget is set to the total number of states, but it is never allowed to exceed the model's maximum context length. We choose the per-attempt cap so that all attempts can be packed into a single context window, while also reserving space for the inserted RETRY separator tokens between attempts. An attempt terminates if it reaches the absorbing terminal state, emits RETRY/PAD/EOS, outputs an out-of-range state id (invalid), or hits the per-attempt token cap.

**Optimization.** We train with AdamW at learning rate $10^{-6}$ with a KL penalty coefficient of 0.001, a clip ratio $\epsilon = 0.2$, and no entropy bonus. We use a train batch size of 256, PPO mini-batch size 64, and PPO micro-batch size 8 per GPU with dynamic batching. We train for 2 epochs on MATH and 1 epoch on Maze. We used 2× NVIDIA L40S or A100 GPUs during the training.

For Markov chain setting, we use AdamW with parameters $(0.9, 0.95)$, PPO-style ratio clipping with $\epsilon = 0.2$. We use learning rate $10^{-5}$ and a KL penalty coefficient of 0.02. For non-trap runs: We train for 1600 steps, batch size of 8, rollout group size $N = 40$, and $K = 4$ attempts per start state. Trap runs: We train for 5000 steps, batch size of 16, rollout group size $N = 40$, and $K = 2$ attempts per start state. We evaluate using $500 \times n_{\text{states}}$ start states and evaluate every 40 iterations.

---

**Verification feedback in MATH**

```
Your previous attempt(s) were incorrect.  Review them and try again.
Solve the problem again from scratch, correcting any mistakes.
Final answer must be \boxed{<answer>}.
```

---

**Verification feedback in Maze**

```
Your previous attempt(s) were incorrect.  Review them and try again.
Solve the maze again from scratch, correcting any mistakes.
Final answer must be a move string over U,D,L,R only in <answer>...</answer>.
```

---

**Example Answer**

```
                            <answer>UU</answer>
```

---

**Example Maze Question**

```
You need to solve the following maze.

'*' denotes the wall that you cannot walk through, '.'  denotes available area that you can walk
through.  'S' denotes the starting point, 'E' denotes the destination.

You need to start from the starting point and cross through the available area to reach the
destination.  There are four movement actions, including Left, Right, Up, Down.

Use L to denote Left movement, R to denote Right movement, U to denote Up movement, and D to
denote Down movement.

You can analyze the maze to find the correct path, and you should write the final path in the
<answer></answer>, e.g., <answer>LLRRDUL</answer>.

## Maze
*********
*.....*E*
*****.*.*
*...*..S*
*.*****.*
*.......*
*.*******
*.......*
*********
Now try to analyze the maze and put the final path in the <answer></answer>.
```

---

**Markov Chain reward and termination logic**

```
For each attempt, the model autoregressively emits state-id tokens.
Reward=1 iff:  (1) the terminal absorbing state is reached; (2) all transitions follow the
allowed adjacency; (3) the number of steps equals the shortest-path distance from the start.
Trap variant:  a trap state is sampled per rollout from two pre-terminal states in the last hub;
visiting it makes Reward=0, and shortest paths are computed with the trap removed.
```

---

**Example Trajectories for Markov Chain for Ver@2**

Example parameters:  $n_{\text{hubs}} = 5$, $m = 6$ ($n_{\text{states}} = 30$), start $s_0 = 0$, terminal=29.
Attempt 1 (failure, backtrack):    $0 \rightarrow 2 \rightarrow 0 \rightarrow 1 \rightarrow \ldots$
RETRY
Attempt 2 (success, shortest path):    $0 \rightarrow 1 \rightarrow 3 \rightarrow 5 \rightarrow \ldots \rightarrow 27 \rightarrow 29$.
Trap variant (e.g., trap=28):  the above success remains valid; any visit to 28 during an
attempt yields $r_k = 0$.

---

## A.5. GRPO Variants

Fix a prompt instance $x$ (and its ground-truth $y$), and draw a GRPO group of $N$ i.i.d. multi-attempt trajectories $\tau_1, \ldots, \tau_N \sim p_\theta(\cdot \mid x)$, as in Section 3. Each trajectory $\tau_n$ executes attempts $k \in \{1, \ldots, K\}$ with early stopping: at attempt $k$ the policy

produces an output $y_{n,k} \sim \pi_\theta(\cdot \mid s_{n,k-1})$ and a deterministic verifier returns $r_{n,k} \in \{0, 1\}$. The stopping time is

$$T_n := \min\{k \le K : r_{n,k} = 1\} \quad \text{(or } T_n = K \text{ if no success).}$$

Define the reach indicator $\mathbf{1}[T_n \ge k]$ and the reached-set

$$S_k := \{n \in [N] : T_n \ge k\}.$$

We also define a token mask $m_{n,t} \in \{0, 1\}$ indicating model-generated tokens, and let $\text{turn}(n, t) \in \{1, \ldots, K\}$ denote the attempt index of token $t$. We use the standard GRPO normalization for all variants below, by dividing centered rewards by the corresponding within-group empirical standard deviation. This rescaling is not part of the RLOO derivations in Sections 3 and 5, but we use it in practice because it improves optimization, and Figure 15 reports direct ablations for this normalization.

**Trajectory-Level GRPO (TL-GRPO).** Let the trajectory-level Ver@K reward be

$$r_n^{\text{TL}} := r_{n,T_n} = \mathbf{1}\Big[ \max_{k \le K} r_{n,k} = 1 \Big].$$

We compute the usual GRPO group-normalized advantage

$$a_n^{\text{TL}} = \frac{r_n^{\text{TL}} - \mu}{\sigma}, \qquad \mu = \frac{1}{N} \sum_{n=1}^{N} r_n^{\text{TL}}, \quad \sigma = \text{Std}(\{r_n^{\text{TL}}\}_{n=1}^N),$$

and assign token-level advantages

$$A_{n,t} = a_n^{\text{TL}} \cdot m_{n,t}.$$

**Naive Attempt-Level GRPO (NAL-GRPO).** For each attempt $k$, we normalize the per-attempt verifier outcomes $r_{n,k}$ *only over trajectories that reach attempt $k$*:

$$a_{n,k}^{\text{NAL}} = \frac{r_{n,k} - \mu_k}{\sigma_k} \quad \text{for } n \in S_k, \qquad \mu_k = \frac{1}{|S_k|} \sum_{n \in S_k} r_{n,k}, \quad \sigma_k = \text{Std}(\{r_{n,k}\}_{n \in S_k}).$$

Each token inherits the attempt-level advantage of its attempt:

$$A_{n,t} = a_{n,\text{turn}(n,t)}^{\text{NAL}} \cdot m_{n,t}.$$

**Calibrated Attempt-Level GRPO (CAL-GRPO).** CAL-GRPO starts from NAL-GRPO and reweights attempts by a future-only factor that approximates each attempt's marginal contribution to final Ver@K success. Define the empirical per-attempt success rates

$$\hat{p}_k := \frac{1}{|S_k|} \sum_{n \in S_k} r_{n,k}.$$

Then define future-only weights:

$$w_k = \prod_{j=k+1}^{K} (1 - \hat{p}_j), \qquad w_K = 1.$$

We renormalize weights within the group to keep update scales comparable, e.g.,

$$\bar{w} = \frac{1}{|\mathcal{K}|} \sum_{k \in \mathcal{K}} w_k, \quad \mathcal{K} := \{k \in [K] : |S_k| > 0\}, \qquad \tilde{w}_k = \frac{w_k}{\bar{w}}.$$

Final token-level advantages are

$$A_{n,t} = \tilde{w}_{\text{turn}(n,t)} a_{n,\text{turn}(n,t)}^{\text{NAL}} m_{n,t}.$$

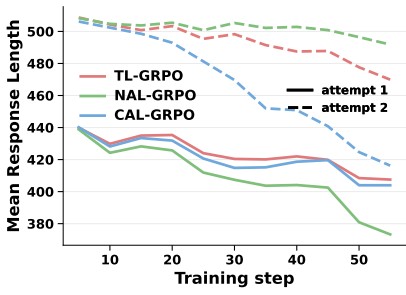

*(a)* **Mean response length** per attempt vs. training step (baselines labeled in legend: TL-GRPO, NAL-GRPO, CAL-GRPO).

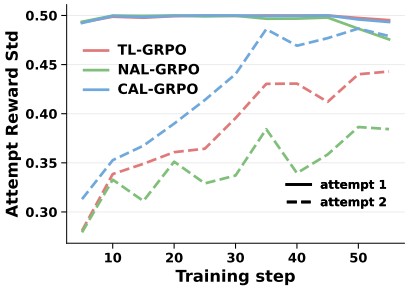

*(b)* **Attempt reward standard deviation** within each GRPO group (signal magnitude for normalization), shown separately for attempts 1 and 2.

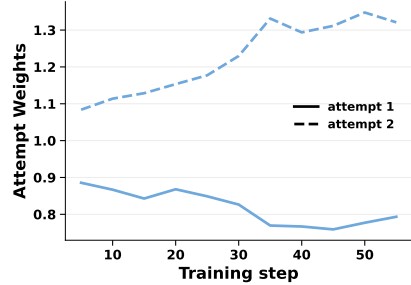

*(c)* **Attempt weights** used by CAL-GRPO (future-only reweighting), plotted for attempts 1 and 2 over training.

*Figure 5.* **Additional metrics for Ver@2 training on MATH.** (a) Response length dynamics per attempt. (b) Per-attempt reward variability used by normalization. (c) Learned future-only attempt reweighting in CAL-GRPO.

## B. Additional Experimental Results for Section 4

**Discussion on the additional results.** Figures 5, 6, 8, 10, 12, 14 and 15 report additional diagnostics and ablations for the three estimators (TL-GRPO/NAL-GRPO/CAL-GRPO). For Maze, we include Ver@2 results on 5×5 and 9×9 mazes (see Figure 7), as well as Ver@4 results on 7×7 and 9×9 mazes (see Figures 9 and 11), and rollout-group-size and standard-deviation-normalization ablations under Ver@2 (see Figures 14 and 15). We make the following observations:

**Weights track marginal contribution.** Across tasks, the future-dependent weights in CAL-GRPO shift learning signal away from early attempts and toward later attempts as training progresses. For Ver@2, CAL-GRPO downweights attempt 1 and upweights attempt 2 over training (Figures 5c, 6c and 8c). For Ver@4, this pattern generalizes: attempts 1–2 are increasingly downweighted while attempts 3–4 are upweighted (Figures 10c and 12c), which counteracts the fact that later attempts are reached on a shrinking subset of hard failures and would otherwise receive a vanishing share of updates. The reweighting is strongest in easier/near-saturating regimes—notably 5×5 under Ver@2 and 7×7 under Ver@4—where retries become highly reliable and the marginal value of further improving the first attempt diminishes. On 9×9 (both Ver@2 and Ver@4), weights remain less extreme, consistent with a harder regime where multiple attempts continue to contribute meaningfully to final Ver@K accuracy (Figures 7a and 9a).

**Gains without longer generations.** Mean response length decreases over training on MATH and across all Maze settings we report, for both Ver@2 and Ver@4 (Figures 5a, 6a, 8a, 10a and 12a), indicating that Ver@K gains are not driven by longer generations. On 5×5 (Ver@2), CAL-GRPO notably shortens the second attempt late in training, directly reducing test-time compute under early stopping. On 9×9 (Ver@2), the dominant effect is a steady reduction in attempt-1 length with controlled retry length, suggesting improved first-pass planning without incurring longer corrections. For Ver@4 on 7×7 and 9×9, response lengths generally drop across attempts 1-4, with later attempts often shortening substantially late in training (Figures 10a and 12a), consistent with learning more efficient correction rather than longer retries.

**Attempt-dependent normalization matters.** The within-group reward standard deviation differs across attempts and evolves over training for both Ver@2 and Ver@4 (Figures 5b, 6b, 8b, 10b and 12b), reinforcing the claim that a single trajectory-level normalization can mis-scale gradients for later attempts. This heterogeneity becomes more salient as $K$ grows: under Ver@4, attempts 3–4 can exhibit markedly different reward-variance dynamics than attempts 1–2. On 5×5 with Ver@2, attempt-2 variability rises early but collapses late as retry outcomes become near-deterministic once performance approaches ceiling; by contrast, on 9×9 with Ver@2 the retry variance remains substantial throughout training, supporting the fact that Ver@2 does not fully saturate and learning signal on retries remains informative (Figure 7a). Similarly, for Ver@4, variance patterns differ across 7×7 vs. 9×9 (Figures 10b and 12b), aligning with the corresponding Ver@4 learning curves (Figures 9a and 11a). Overall, these demonstrations help explain why CAL-GRPO's reweighting becomes increasingly important for larger $K$ as it preserves learning pressure on later-attempt correction when those retries remain a key driver of marginal Ver@K improvements.

**Variance reduction estimates.** Figure 13 estimates the bias and variance of the gradient estimators at fixed checkpoints of the trap Markov-chain Ver@2 experiment. For clarity, we measure the score-function part of the GRPO update, i.e., the part

affected by the TL/NAL/CAL advantage assignment, and omit PPO clipping and KL regularization from this diagnostic. Let

$$R(\tau) := \mathbf{1}\{\max_{i \leq K} r_i = 1\}$$

denote the Ver@K reward of a trajectory, and define the true Ver@K policy gradient

$$g_\star(\theta) := \nabla_\theta J(\theta) = \nabla_\theta \mathbb{E}_{(x,y)\sim\mathcal{D}} \mathbb{E}_{\tau \sim p_\theta(\cdot|x)} [R(\tau)].$$

By the log-derivative identity,

$$g_\star(\theta) = \mathbb{E}_{(x,y),\tau} \left[ R(\tau) \sum_{i=1}^{K} \mathbf{1}\{T(\tau) \geq i\} \nabla_\theta \log \pi_\theta(y_i \mid s_{i-1}) \right].$$

For a rollout group $\mathcal{G} = \{\tau_n\}_{n=1}^{N}$, write

$$h_{n,i} := \mathbf{1}\{T_n \geq i\} \nabla_\theta \log \pi_\theta(y_{n,i} \mid s_{n,i-1}), \qquad R_n := R(\tau_n), \qquad S_i := \{n : T_n \geq i\}.$$

Each method $M \in \{\text{TL}, \text{NAL}, \text{CAL}\}$ defines a gradient estimator of the form

$$\widehat{g}_M(\theta; \mathcal{G}) = \frac{1}{N} \sum_{n=1}^{N} \sum_{i=1}^{K} a_{n,i}^M h_{n,i},$$

where the only difference across methods is the advantage coefficient $a_{n,i}^M$. For TL-GRPO,

$$a_{n,i}^{\text{TL}} = \frac{R_n - \mu^{\text{TL}}}{\sigma^{\text{TL}} + \epsilon}, \qquad \mu^{\text{TL}} = \frac{1}{N} \sum_{m=1}^{N} R_m, \qquad \sigma^{\text{TL}} = \text{Std}\{R_m\}_{m=1}^{N}.$$

For NAL-GRPO, rewards are normalized separately at each attempt:

$$a_{n,i}^{\text{NAL}} = \frac{r_{n,i} - \mu_i}{\sigma_i + \epsilon}, \qquad n \in S_i,$$

where

$$\mu_i = \frac{1}{|S_i|} \sum_{m \in S_i} r_{m,i}, \qquad \sigma_i = \text{Std}\{r_{m,i} : m \in S_i\}.$$

For CAL-GRPO, the NAL advantage is reweighted by the empirical future-failure probability:

$$a_{n,i}^{\text{CAL}} = \widetilde{w}_i a_{n,i}^{\text{NAL}}, \qquad w_i = \prod_{j=i+1}^{K} (1 - \widehat{p}_j), \qquad \widehat{p}_j = \frac{1}{|S_j|} \sum_{m \in S_j} r_{m,j},$$

with the normalized weight

$$\widetilde{w}_i = \frac{w_i}{|\mathcal{K}|^{-1} \sum_{\ell \in \mathcal{K}} w_\ell}, \qquad \mathcal{K} := \{i : |S_i| > 0\}.$$

The population bias and variance of method $M$ are

$$\text{Bias}_M(\theta) := \mathbb{E}_{\mathcal{G}} [\widehat{g}_M(\theta; \mathcal{G})] - g_\star(\theta),$$

$$\mathcal{B}_M^2(\theta) := \|\text{Bias}_M(\theta)\|_2^2,$$

and

$$\mathcal{V}_M(\theta) := \mathbb{E}_{\mathcal{G}} \left[ \left\| \widehat{g}_M(\theta; \mathcal{G}) - \mathbb{E}_{\mathcal{G}} \widehat{g}_M(\theta; \mathcal{G}) \right\|_2^2 \right] = \text{tr} \, \text{Cov}_{\mathcal{G}} (\widehat{g}_M(\theta; \mathcal{G})).$$

Since $g_\star(\theta)$ is not available exactly for the neural policy, we approximate it with a high-sample Monte Carlo reference estimator

$$\widehat{g}_\star = \frac{1}{L_{\text{ref}} N_{\text{ref}}} \sum_{\ell=1}^{L_{\text{ref}}} \sum_{n=1}^{N_{\text{ref}}} R_n^{(\ell)} \sum_{i=1}^{K} h_{n,i}^{(\ell)},$$

using many more rollouts than the diagnostic estimators. Then, for each method $M$, we draw $L$ independent rollout groups and compute

$$\bar{g}_M = \frac{1}{L} \sum_{\ell=1}^{L} \widehat{g}_M^{(\ell)}.$$

The quantities plotted in Figure 13 are

$$\widehat{\mathcal{B}}_M^2 = \left\| \bar{g}_M - \widehat{g}_\star \right\|_2^2$$

and

$$\widehat{\mathcal{V}}_M = \frac{1}{L-1} \sum_{\ell=1}^{L} \left\| \widehat{g}_M^{(\ell)} - \bar{g}_M \right\|_2^2.$$

For the attempt-wise variance plots, we apply the same calculation to the attempt-specific component

$$\widehat{g}_{M,i}(\theta; \mathcal{G}) = \frac{1}{N} \sum_{n=1}^{N} a_{n,i}^M h_{n,i}.$$

This diagnostic also clarifies the source of the NAL bias. Under Assumption 5.1, let

$$\rho_i := \mathbb{P}(r_i = 1 \mid T \geq i)$$

be the success probability of attempt $i$ conditional on reaching it. Then

$$J(\theta) = 1 - \prod_{i=1}^{K}(1 - \rho_i),$$

so the true gradient can be written as

$$g_\star(\theta) = \sum_{i=1}^{K} \left( \prod_{j<i}(1 - \rho_j) \right) \left( \prod_{j>i}(1 - \rho_j) \right) \nabla_\theta \rho_i.$$

The ideal TL and CAL estimators target this gradient, whereas the ideal NAL estimator replaces the future-failure factor by one:

$$\mathbb{E}[\widehat{g}_{\text{NAL}}] = \sum_{i=1}^{K} \left( \prod_{j<i}(1 - \rho_j) \right) \nabla_\theta \rho_i.$$

Thus the NAL bias relative to the Ver@K gradient is

$$\text{Bias}_{\text{NAL}} = \sum_{i=1}^{K} \left( \prod_{j<i}(1 - \rho_j) \right) \left[ 1 - \prod_{j>i}(1 - \rho_j) \right] \nabla_\theta \rho_i,$$

which overweights earlier attempts whenever later attempts still have nonzero probability of success. CAL-GRPO corrects this mismatch by multiplying attempt $i$ by an estimate of the future-failure probability $\prod_{j>i}(1 - \rho_j)$, while TL-GRPO remains unbiased but higher-variance because it uses the terminal reward for all attempts.

**Ablation on standard deviation normalization.** Consistent with the attempt-dependent variance patterns above, the direct ablations in Figure 15 show that dividing by the within-group empirical standard deviation improves sample efficiency for TL-GRPO, NAL-GRPO, and CAL-GRPO across the no-trap Markov chain and Maze settings. Although our theoretical derivations are written in an RLOO-style form without this GRPO rescaling, these empirical results motivate our use of the standard GRPO normalization in all experiments.

**Ablation on group size.** CAL-GRPO estimates its weights directly from the current rollout group. For Ver@2, the quantity is $\hat{p}_2 = \frac{1}{|S_2|} \sum_{n \in S_2} r_{n,2}$, which is estimated from the trajectories in the current group that reach attempt 2, and the calibrated weights reduce to $w_1 = 1 - \hat{p}_2$ and $w_2 = 1$. Hence, the quality of the calibration depends on how well the current rollouts estimate second-attempt success. Figure 14 compares rollout group sizes across the no-trap Markov chain and Maze settings. On Maze Ver@2, increasing the rollout group size from $N=10$ to $N=16$ improves sample efficiency for all three methods, with the clearest early gain for CAL-GRPO. On the no-trap Markov chain Ver@4 task, larger groups similarly improve optimization stability and sample efficiency. Larger groups provide less noisy within-group normalization and, for CAL-GRPO, more reliable empirical estimates of the future-attempt success probabilities used in the calibrated weights. These results motivate our use of moderately large rollout groups for CAL-GRPO throughout our experiments.

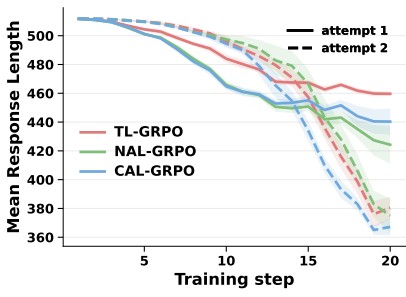 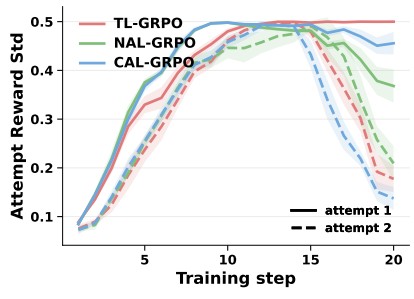 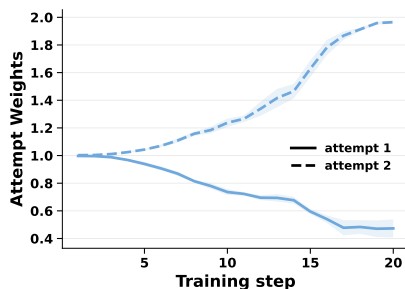

*(a)* **Mean response length** per attempt vs. training step (baselines labeled in legend: TL-GRPO, NAL-GRPO, CAL-GRPO).

*(b)* **Attempt reward standard deviation** within each GRPO group, shown separately for attempts 1 and 2.

*(c)* **Attempt weights** used by CAL-GRPO (future-only reweighting), for attempts 1 and 2 over training.

*Figure 6.* **Additional metrics for Ver@2 training on** 5×5 **Maze task.** (a) Response length dynamics per attempt. (b) Per-attempt reward variation used during the per-attempt normalization. (c) Learned future-only attempt reweighting in CAL-GRPO.

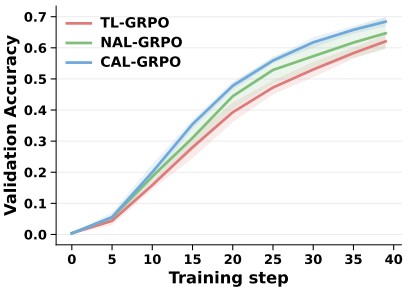 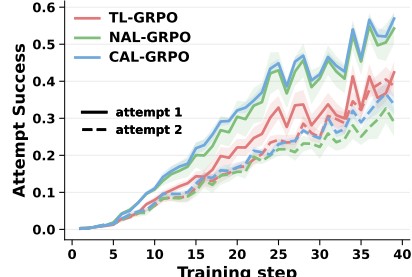 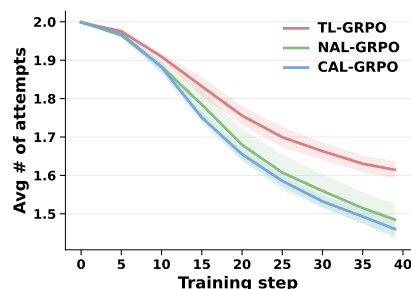

*(a)* **Ver@2 validation accuracy on 9×9 Maze** across training; CAL-GRPO achieves the highest Ver@2, with NAL-GRPO improving over TL-GRPO.

*(b)* **Attempt-wise success.** Solid: attempt-1 success. Dashed: attempt-2 success given attempt-1 failure; NAL-GRPO boost attempt-1 success while CAL improves correction.

*(c)* **Average attempts** Ver@2; NAL/CAL reduce number of attempts relative to TL-GRPO.

*Figure 7.* **Optimizing the Ver@2 objective on 9×9 Maze with GRPO variants.** We compare Trajectory-Level GRPO (TL-GRPO), Naive Attempt-Level GRPO (NAL-GRPO), and Calibrated Attempt-Level GRPO (CAL-GRPO). **(a)** CAL-GRPO attains the best Ver@2. **(b)** Improvements mainly come from higher attempt-1 success with competitive attempt-2 correction. **(c)** NAL/CAL reduce the average number of attempts compared to TL-GRPO.

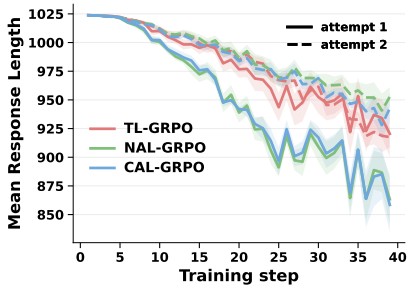 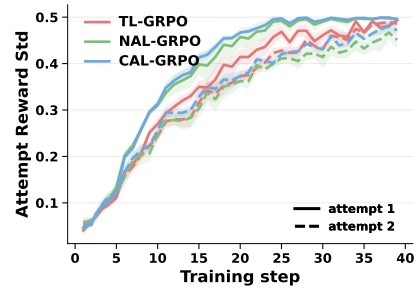 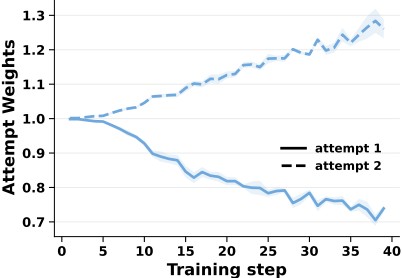

*(a)* **Mean response length** per attempt vs. training step (baselines labeled in legend: TL-GRPO, NAL-GRPO, CAL-GRPO).

*(b)* **Attempt reward standard deviation** within each GRPO group, shown separately for attempts 1 and 2.

*(c)* **Attempt weights** used by CAL-GRPO (future-only reweighting), for attempts 1 and 2 over training.

*Figure 8.* **Additional metrics for Ver@2 training on 9×9 Maze task.** (a) Response length dynamics per attempt. (b) Per-attempt reward variation used during the per-attempt normalization. (c) Learned future-only attempt reweighting in CAL-GRPO.

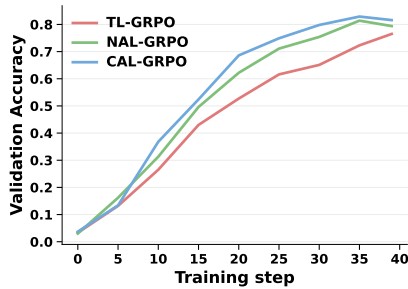
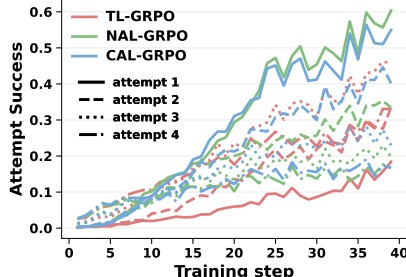
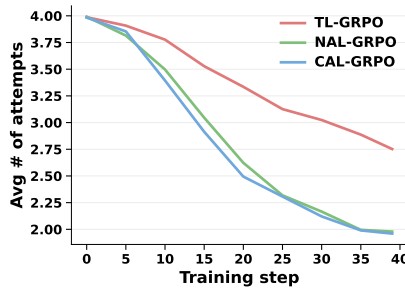

*(a)* **Ver@4 validation accuracy on 9×9 Maze** across training; CAL-GRPO achieves the highest Ver@4, with NAL-GRPO improving over TL-GRPO.

*(b)* **Attempt-wise success.** Curves show success on attempt *i* conditioned on failing all earlier attempts. NAL-GRPO boosts early-attempt success while CAL-GRPO improves later-attempt correction.

*(c)* **Average attempts** under Ver@4; NAL/-CAL reduce the number of attempts relative to TL-GRPO.

*Figure 9.* **Optimizing the Ver@4 objective on 9×9 Maze with GRPO variants.** We compare Trajectory-Level GRPO (TL-GRPO), Naive Attempt-Level GRPO (NAL-GRPO), and Calibrated Attempt-Level GRPO (CAL-GRPO). **(a)** CAL-GRPO attains the best Ver@4. **(b)** Improvements reflect stronger attempt-wise performance, with CAL-GRPO emphasizing correction on later attempts. **(c)** NAL/CAL reduce the average number of attempts compared to TL-GRPO.

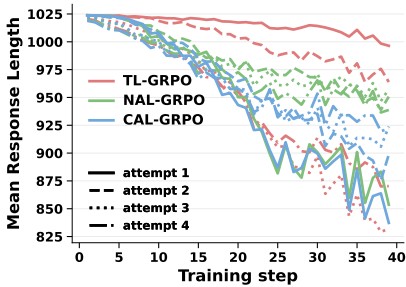
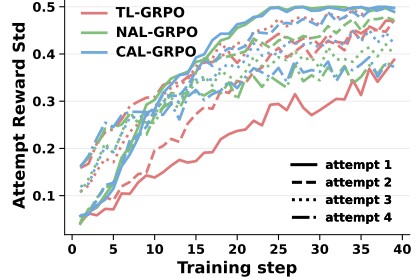
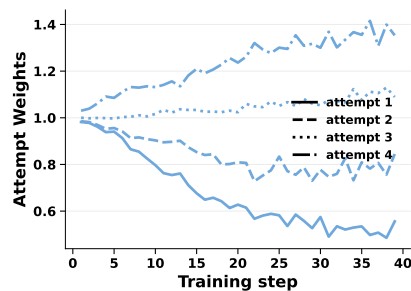

*(a)* **Mean response length** per attempt vs. training step (baselines labeled in legend: TL-GRPO, NAL-GRPO, CAL-GRPO).

*(b)* **Attempt reward standard deviation** within each GRPO group, shown separately for attempts 1–4.

*(c)* **Attempt weights** used by CAL-GRPO (future-only reweighting), for attempts 1–4 over training.

*Figure 10.* **Additional metrics for Ver@4 training on 9×9 Maze task.** (a) Response length dynamics per attempt. (b) Per-attempt reward variation used during the per-attempt normalization. (c) Learned future-only attempt reweighting in CAL-GRPO.

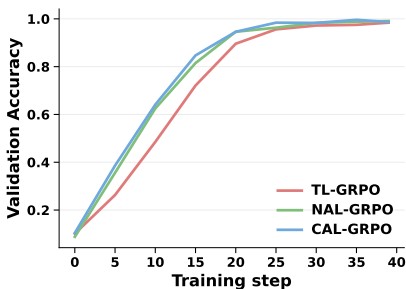
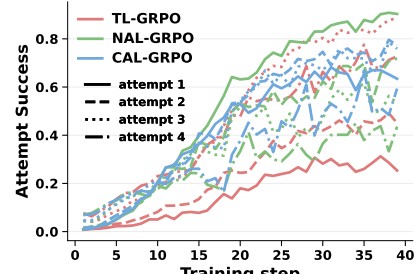
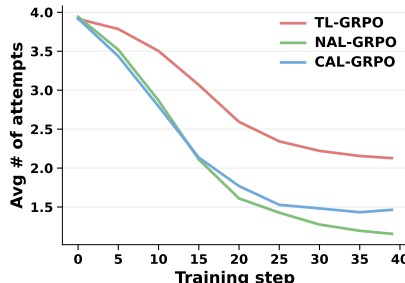

*(a)* **Ver@4 validation accuracy on 7×7 Maze** across training; CAL-GRPO achieves the highest Ver@4, with NAL-GRPO improving over TL-GRPO.

*(b)* **Attempt-wise success.** Line styles denote attempts 1–4. Curves show success on attempt *i* given all earlier attempts failed; NAL-GRPO boosts early success while CAL-GRPO improves correction.

*(c)* **Average attempts** under Ver@4; NAL/-CAL reduce the number of attempts relative to TL-GRPO.

*Figure 11.* **Optimizing the Ver@4 objective on 7×7 Maze with GRPO variants.** We compare Trajectory-Level GRPO (TL-GRPO), Naive Attempt-Level GRPO (NAL-GRPO), and Calibrated Attempt-Level GRPO (CAL-GRPO). **(a)** CAL-GRPO attains the best Ver@4. **(b)** Improvements reflect stronger attempt-wise performance, with CAL-GRPO emphasizing correction on later attempts. **(c)** NAL/CAL reduce the average number of attempts compared to TL-GRPO.

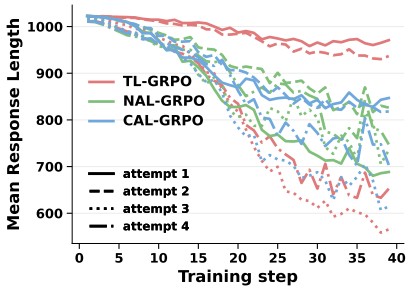

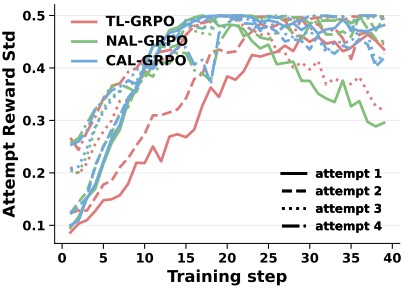

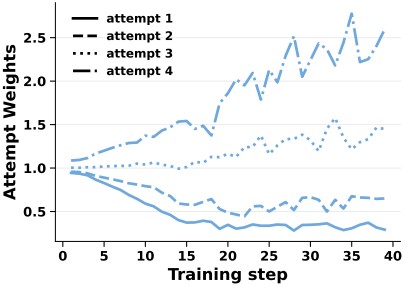

*(a)* **Mean response length** per attempt vs. training step (baselines labeled in legend: TL-GRPO, NAL-GRPO, CAL-GRPO).

*(b)* **Attempt reward standard deviation** within each GRPO group, shown separately for attempts 1–4.

*(c)* **Attempt weights** used by CAL-GRPO (future-only reweighting), for attempts 1–4 over training.

*Figure 12.* **Additional metrics for Ver@4 training on 7×7 Maze task.** (a) Response length dynamics per attempt. (b) Per-attempt reward variation used during the per-attempt normalization. (c) Learned future-only attempt reweighting in CAL-GRPO.

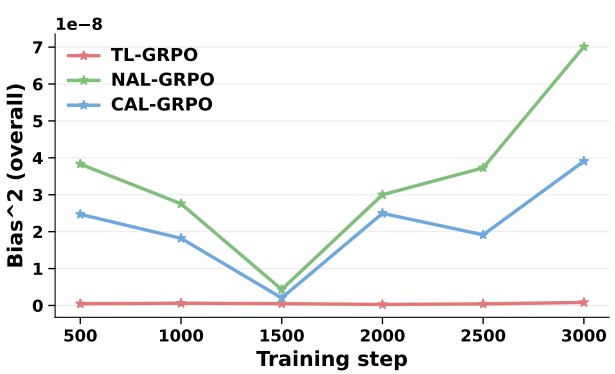

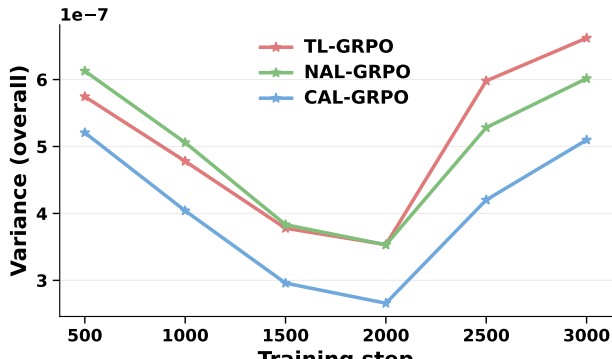

*(a)* **Trap Markov chain Ver@2** ($n_{\text{hubs}}$=5, $m$=6). Illustration of $(\text{bias})^2$ vs training step for the whole trajectory's gradient across different baselines.

*(b)* **Trap Markov chain Ver@2** ($n_{\text{hubs}}$=5, $m$=6). Illustration of variance vs training step for the whole trajectory's gradient across different baselines.

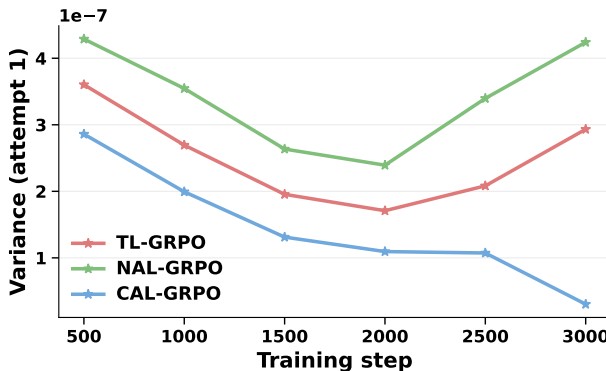

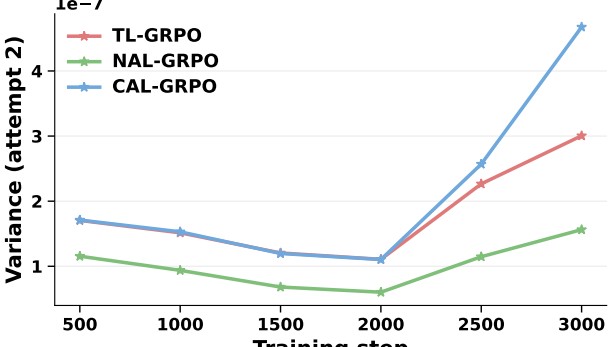

*(c)* **Trap Markov chain Ver@2** ($n_{\text{hubs}}$=5, $m$=6). Illustration of variance vs training step for the first attempt's gradient across different baselines.

*(d)* **Trap Markov chain Ver@2** ($n_{\text{hubs}}$=5, $m$=6). Illustration of variance vs training step for the second attempt's gradient given the first attempt across different baselines.

*Figure 13.* **Practical Demonstration of Unbiasedness and Variance Reduction on the Trap Markov chain Ver@2.**

# C. Proofs Section 3

*Proof.* Fix a problem instance $(x, y)$ and suppress its dependence in the notation. Let $\tau$ be a rollout with stopping time $T \in [K]$ and terminal Ver@K reward

$$R := r_T \in \{0, 1\}.$$

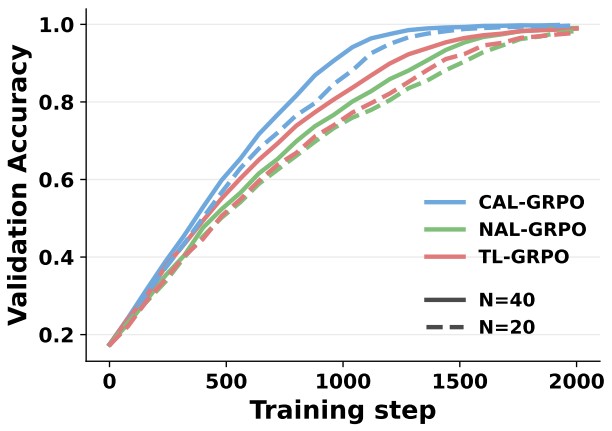

*(a)* **No-trap Markov chain Ver@4.** $n_{hubs}$=5, $m$=6; rollout group sizes $N$=40 vs. $N$=20.

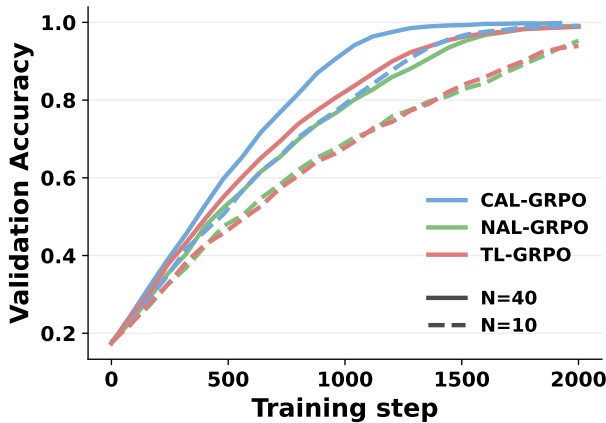

*(b)* **No-trap Markov chain Ver@4.** $n_{hubs}$=5, $m$=6; rollout group sizes $N$=40 vs. $N$=10.

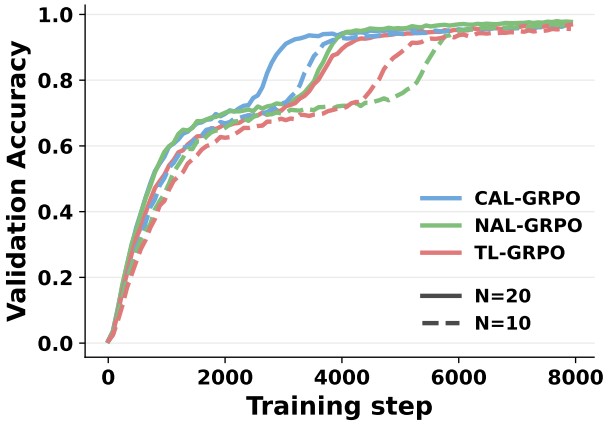

*(c)* **Trap Markov chain Ver@2.** $n_{hubs}$=5, $m$=6; rollout group sizes $N$=20 vs. $N$=10.

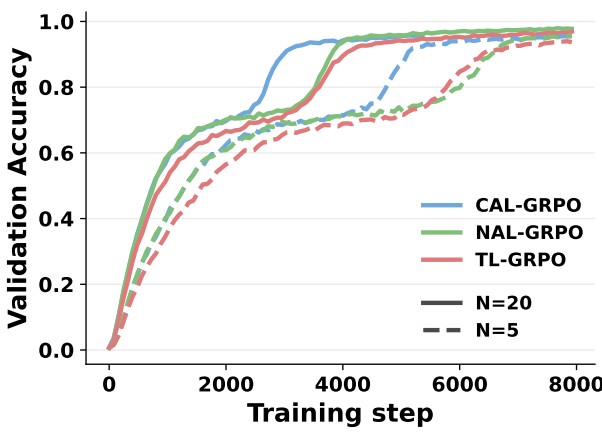

*(d)* **Trap Markov chain Ver@2.** $n_{hubs}$=5, $m$=6; rollout group sizes $N$=20 vs. $N$=5.

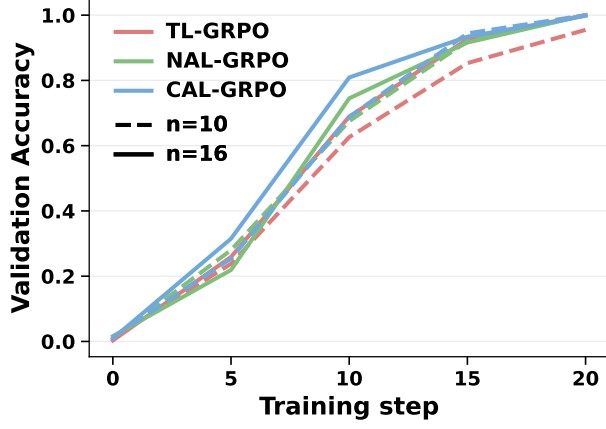

*(e)* 5×5 **Maze Ver@2.** Rollout group sizes $N$=16 vs. $N$=10.

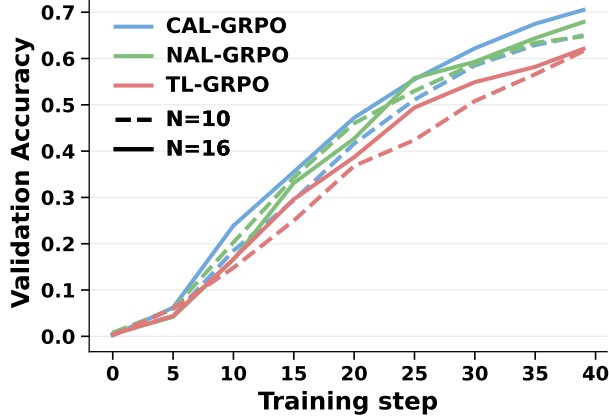

*(f)* 9×9 **Maze Ver@2.** Rollout group sizes $N$=16 vs. $N$=10.

*Figure 14.* **Ablations on rollout group size for no-trap Markov chain Ver@4, Trap Markov chain Ver@2, and Maze Ver@2.** We compare rollout group sizes for TL-GRPO, NAL-GRPO, and CAL-GRPO on the no-trap Markov chain task, the trap Markov chain task, and on 5×5 and 9×9 Maze tasks. Across these settings, larger rollout groups improve sample efficiency, with the clearest early gains for CAL-GRPO.

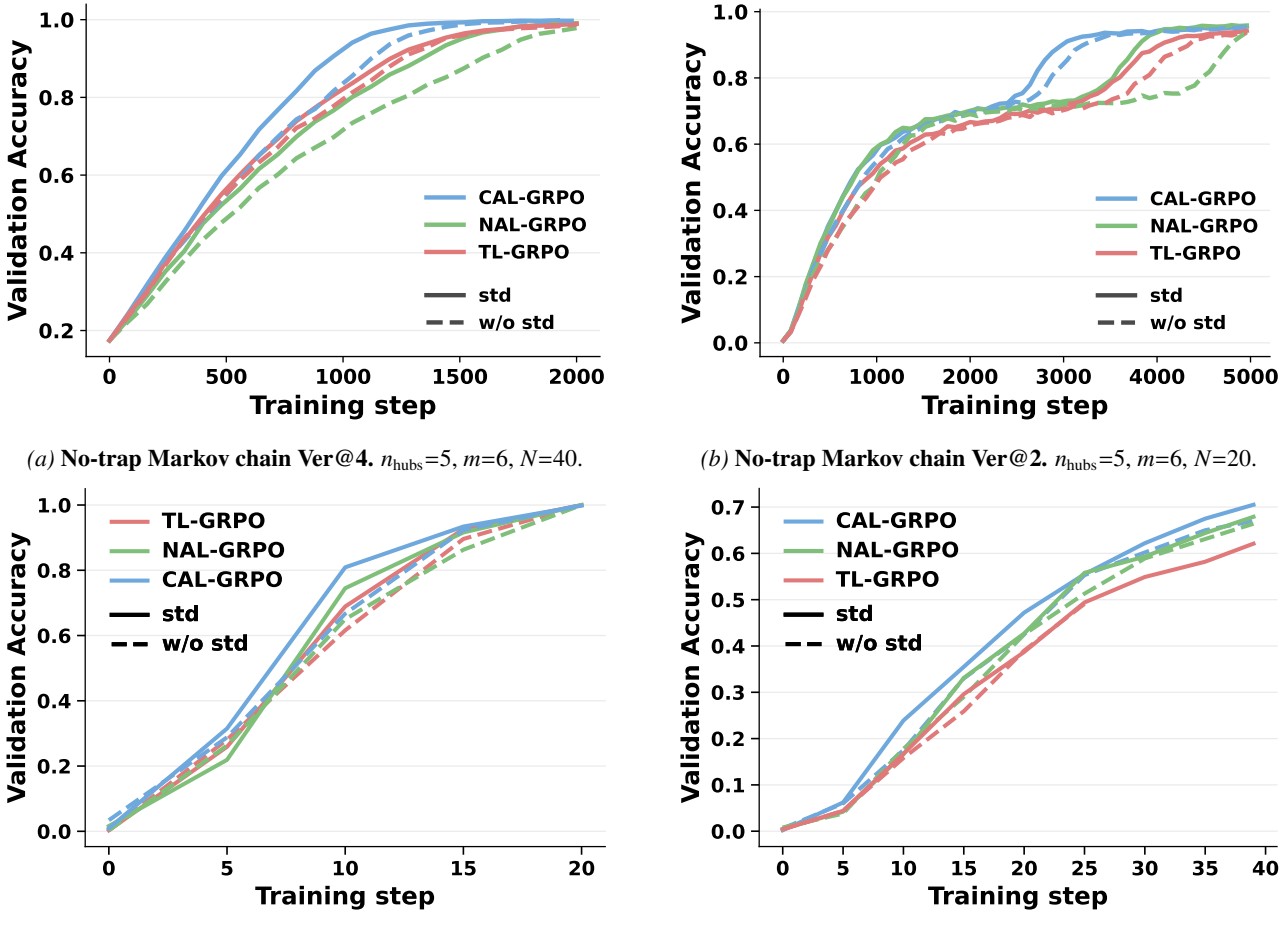

*(a)* **No-trap Markov chain Ver@4.** $n_{\text{hubs}}=5$, $m=6$, $N=40$.

*(b)* **No-trap Markov chain Ver@2.** $n_{\text{hubs}}=5$, $m=6$, $N=20$.

*(c)* 5×5 **Maze Ver@2.** Standard-deviation normalization ablation at $N=16$.

*(d)* 9×9 **Maze Ver@2.** Standard-deviation normalization ablation at $N=16$.

*Figure 15.* **Ablations on within-group standard-deviation normalization for no-trap Markov chain Ver@4, trap Markov chain Ver@2, and Maze Ver@2.** Solid curves use the standard GRPO normalization with within-group mean centering and standard-deviation scaling; dashed curves use mean-only normalization without standard-deviation scaling. We compare TL-GRPO, NAL-GRPO, and CAL-GRPO on the no-trap and trap Markov chain task; and on 5×5 and 9×9 Maze tasks. Standard-deviation scaling improves sample efficiency and is used in our main experiments.

For attempt $i$, define the score

$$g_i := \mathbf{1}\{T \geq i\}\nabla_\theta \log \pi_\theta(y_i \mid s_{i-1}).$$

Let $\mathcal{F}_i$ denote the sigma-field generated by the trajectory up to and including attempt $i$, and define the Doob conditional success probability

$$Q_i := \mathbb{E}[R \mid \mathcal{F}_i].$$

By the deterministic verifier assumption, on the event $\{T \geq i\}$ we have

$$Q_i = r_i + (1 - r_i)V_{\theta,i}(s_i),$$

which is exactly the attempt-level reward.

Now draw $N \geq 2$ i.i.d. trajectories $\tau_1, \ldots, \tau_N \sim p_\theta(\cdot \mid x)$. Let

$$S_i := \{n \in [N] : T_n \geq i\}, \qquad M_i := |S_i|.$$

For simplicity, condition on the event $M_i \geq 2$; if $M_i = 1$, the corresponding leave-one-out term is defined as zero, as in the algorithm.

For the per-attempt comparison, define the reached-set leave-one-out advantages

$$A_{n,i}^{\text{TL}} := R_n - \frac{1}{M_i - 1} \sum_{\substack{m \in S_i \\ m \neq n}} R_m,$$

and

$$A_{n,i}^{\text{AL}} := Q_{n,i} - \frac{1}{M_i - 1} \sum_{\substack{m \in S_i \\ m \neq n}} Q_{m,i}.$$

The corresponding per-attempt gradient terms are

$$Z_i^{\text{TL}} := \frac{1}{N} \sum_{n \in S_i} A_{n,i}^{\text{TL}} g_{n,i}, \qquad Z_i^{\text{AL}} := \frac{1}{N} \sum_{n \in S_i} A_{n,i}^{\text{AL}} g_{n,i}.$$

We first show unbiasedness. Since $Q_i = \mathbb{E}[R \mid \mathcal{F}_i]$ and $g_i$ is $\mathcal{F}_i$-measurable,

$$\mathbb{E}[Rg_i] = \mathbb{E}[\mathbb{E}[Rg_i \mid \mathcal{F}_i]] = \mathbb{E}[Q_i g_i].$$

Therefore

$$\sum_{i=1}^{K} \mathbb{E}[Q_i g_i] = \sum_{i=1}^{K} \mathbb{E}[R g_i] = \mathbb{E}\left[ R \sum_{i=1}^{K} g_i \right].$$

Because

$$\sum_{i=1}^{K} g_i = \nabla_\theta \log p_\theta(\tau \mid x),$$

the log-derivative identity gives

$$\mathbb{E}\left[ R \sum_{i=1}^{K} g_i \right] = \nabla_\theta \mathbb{E}[R] = \nabla_\theta \rho_\theta(x, y).$$

The leave-one-out baselines do not change the expectation: for each $n$, the baseline excludes trajectory $n$ and is independent of the attempt-$i$ score conditional on the pre-attempt state, while

$$\mathbb{E}[\nabla_\theta \log \pi_\theta(y_{n,i} \mid s_{n,i-1}) \mid s_{n,i-1}] = 0.$$

Hence both the TL and AL leave-one-out estimators are unbiased:

$$\mathbb{E}[\widehat{G}_{\text{RLOO}}^{\text{TL}}(\theta; x, y)] = \mathbb{E}[\widehat{G}_{\text{RLOO}}^{\text{AL}}(\theta; x, y)] = \nabla_\theta \rho_\theta(x, y).$$

We now prove the covariance reduction. Let

$$\mathcal{G}_i := \sigma(\mathcal{F}_{1,i}, \ldots, \mathcal{F}_{N,i})$$

be the group filtration up to attempt $i$. Conditional on $\mathcal{G}_i$, the set $S_i$, the scores $g_{n,i}$, and the quantities $Q_{n,i}$ are fixed. Moreover, by independence across trajectories,

$$\mathbb{E}[R_n \mid \mathcal{G}_i] = \mathbb{E}[R_n \mid \mathcal{F}_{n,i}] = Q_{n,i}.$$

Therefore, conditioning $Z_i^{\text{TL}}$ on $\mathcal{G}_i$ replaces every terminal reward $R_n$ by its conditional expectation $Q_{n,i}$:

$$\mathbb{E}[Z_i^{\text{TL}} \mid \mathcal{G}_i] = \frac{1}{N} \sum_{n \in S_i} \left( Q_{n,i} - \frac{1}{M_i - 1} \sum_{\substack{m \in S_i \\ m \neq n}} Q_{m,i} \right) g_{n,i}$$

$$= Z_i^{\text{AL}}.$$

Thus $Z_i^{\text{AL}}$ is the Rao–Blackwellization of $Z_i^{\text{TL}}$ with respect to $\mathcal{G}_i$.

Using the law of total covariance,

$$\text{Cov}(Z_i^{\text{TL}}) = \text{Cov}\big(\mathbb{E}[Z_i^{\text{TL}} \mid \mathcal{G}_i]\big) + \mathbb{E}\big[\text{Cov}(Z_i^{\text{TL}} \mid \mathcal{G}_i)\big].$$

Since $\mathbb{E}[Z_i^{\text{TL}} \mid \mathcal{G}_i] = Z_i^{\text{AL}}$, this becomes

$$\text{Cov}(Z_i^{\text{TL}}) = \text{Cov}(Z_i^{\text{AL}}) + \mathbb{E}\big[\text{Cov}(Z_i^{\text{TL}} \mid \mathcal{G}_i)\big].$$

The second term is positive semidefinite, so

$$\text{Cov}(Z_i^{\text{TL}}) \succeq \text{Cov}(Z_i^{\text{AL}}).$$

Equivalently, for every direction $v$,

$$\text{Var}\big(\langle v, Z_i^{\text{TL}} \rangle\big) \geq \text{Var}\big(\langle v, Z_i^{\text{AL}} \rangle\big).$$

The inequality is strict in any direction $v$ for which

$$\mathbb{E}\big[\text{Var}\big(\langle v, Z_i^{\text{TL}} \rangle \mid \mathcal{G}_i\big)\big] > 0.$$

Ignoring degenerate zero-score cases, this happens whenever the terminal reward still has nontrivial future randomness after attempt $i$, i.e. whenever there are reachable states $s_i$ with

$$0 < V_{\theta,i}(s_i) < 1.$$

If $V_{\theta,i}(s_i) \in \{0, 1\}$ for all reachable $s_i$, then $R = Q_i$ almost surely after conditioning on $\mathcal{F}_i$, so the Rao–Blackwell gap vanishes. $\qquad\square$

### C.1. Noisy Verifier

Under a frozen noisy binary verifier, or a fixed learned verifier, modeled as a stochastic channel

$$q_i(\widetilde{r}_i \mid s_{i-1}, y_i),$$

which is independent of the policy parameters, an analogue of Theorem 3.1 still holds for the noisy Ver@$K$ objective

$$\widetilde{\rho}_\theta(x, y) = \mathbb{E}[\widetilde{r}_{\widetilde{T}}],$$

where $\widetilde{T}$ is the stopping time induced by the noisy verifier. The key observation is that our Appendix C proof is already written in terms of the Doob conditional success probability

$$Q_i = \mathbb{E}[R \mid \mathcal{F}_i].$$

Determinism is only used there to obtain the explicit recursion

$$Q_i = r_i + (1 - r_i)V_{\theta,i}.$$

Therefore, if we replace the reward $R$ by the observed verifier reward $\widetilde{R}$, and replace $Q_i$ by

$$\widetilde{Q}_i = \mathbb{E}[\widetilde{R} \mid \mathcal{F}_i],$$

the same argument yields

$$\mathbb{E}\big[\widehat{G}_{\text{RLOO}}^{\text{TL}}\big] = \mathbb{E}\big[\widehat{G}_{\text{RLOO}}^{\text{AL}}\big] = \nabla_\theta \widetilde{\rho}_\theta(x, y),$$

and the per-attempt variance-reduction guarantee is unchanged.

For CAL, under the same attempt-index-independence assumption used in Theorem 5.2, the substitution $r_i \mapsto \widetilde{r}_i$ also preserves unbiasedness for the noisy-verifier objective, because the Appendix D argument only requires conditional independence of the observed per-attempt outcomes.

Next, we ask whether the noisy verifier induces the same objective as the true verifier, i.e., whether

$$\widetilde{\rho}_\theta(x, y) = \rho_\theta(x, y).$$

In general, it does not, because the verifier controls continuation to the next attempt and therefore changes the effective weighting over attempts. A simple counterexample is $K = 2$: suppose the first attempt is correct with probability $p$, and the verifier has type-I and type-II error rates $q_1$ and $q_2$. Under the true verifier, the expected number of second attempts is $1 - p$, whereas under the noisy verifier it is

$$pq_2 + (1 - p)q_1.$$

Since trajectory-level optimization is unbiased, and it places no additional weighting across attempts, these two verifiers would need to satisfy

$$1 - p = pq_2 + (1 - p)q_1,$$

which implies

$$p = \frac{q_2}{q_1 + q_2}.$$

But $p$ changes throughout training, while $q_1$ and $q_2$ are fixed. Hence this equality cannot hold in general over the course of training, so unbiasedness with respect to the true objective is not preserved globally.

## D. Proofs for Section 5

**Theorem D.1** (CAL is unbiased under attempt-index independence; per-attempt Rao–Blackwell variance reduction). *Fix $(x, y)$ and suppress $(x, y)$ in notation. Assume Assumption 5.1 holds. Let $\rho_\theta(x, y) := \mathbb{E}[R]$ denote the Verification@K success probability.*

*Draw $N \geq 2$ i.i.d. trajectories $\tau_1, \ldots, \tau_N \sim p_\theta(\cdot \mid x)$ and define $\mathcal{S}_i := \{n \in [N] : \mathcal{T}(\tau_n) \geq i\}$ and $M_i := |\mathcal{S}_i|$, assuming $M_i \geq 2$ a.s. For each $n \in \mathcal{S}_i$, define the leave-one-out estimate of the attempt-$i$ success probability*

$$\rho_{n,i} := \frac{1}{M_i - 1} \sum_{\substack{m \in \mathcal{S}_i \\ m \neq n}} r_{m,i}, \qquad A_{n,i}^{\text{CAL}} := r_{n,i} - \rho_{n,i}.$$

*Define the CAL* future-failure *weight*

$$w_{n,i}^{\text{CAL}} := \prod_{j=i+1}^{K} (1 - \rho_{n,j}), \qquad i \in [K], \tag{11}$$

*(where for $j > i$ we interpret $\rho_{n,j}$ as the leave-one-out average over $\mathcal{S}_j$ excluding $n$ if $n \in \mathcal{S}_j$ and otherwise as the full average over $\mathcal{S}_j$; this makes $\rho_{n,j}$ well-defined for all $n$ and $j$ and agrees with the above definition when $n \in \mathcal{S}_j$). Define the per-attempt CAL term*

$$Z_i^{\text{CAL}} := \frac{1}{M_i} \sum_{n \in \mathcal{S}_i} w_{n,i}^{\text{CAL}} A_{n,i}^{\text{CAL}} g_{n,i}, \qquad i \in [K],$$

*and the CAL group estimator (in the same scaling as the TL/AL definitions)*

$$\widehat{G}_{\text{RLOO}}^{\text{CAL}}(\theta; x, y) := \sum_{i=1}^{K} \frac{M_i}{N} Z_i^{\text{CAL}} = \frac{1}{N} \sum_{i=1}^{K} \sum_{n \in \mathcal{S}_i} w_{n,i}^{\text{CAL}} A_{n,i}^{\text{CAL}} g_{n,i}. \tag{12}$$

*Then:*

$$\mathbb{E}\left[\widehat{G}_{\text{RLOO}}^{\text{CAL}}(\theta; x, y)\right] = \nabla_\theta \rho_\theta(x, y). \tag{13}$$

*Proof.* Let $\mathcal{G}_i := \sigma(\mathcal{F}_{1,i}, \ldots, \mathcal{F}_{N,i})$ be the group filtration.

Assumption 5.1 implies that there exist policies $(\pi_\theta^{(i)}(\cdot \mid x))_{i=1}^{K}$ such that $p_\theta(y_i \mid s_{1:i-1}) = \pi_\theta^{(i)}(y_i \mid x)$. Hence, conditional on $x$, the attempt outputs $y_1, \ldots, y_K$ are independent across $i$, and so are the correctness indicators $(r_i)_{i=1}^{K}$.

Define the attempt-$i$ success probability (conditional on reaching attempt $i$)

$$\rho_i := \mathbb{P}(r_i = 1 \mid \mathcal{T} \geq i).$$

Under Assumption 5.1, $\rho_i$ depends only on $(x, i, \theta)$ and not on the content of the prior history.

Let

$$u_i := \mathbb{P}(r_{i+1} = 0, \dots, r_K = 0 \mid \mathcal{T} \geq i) = \prod_{j=i+1}^{K} (1 - \rho_j), \qquad i \in [K],$$

(with $u_K := 1$ by the empty-product convention). Then, on $\{\mathcal{T} \geq i\}$, the Doob conditional success probability satisfies

$$Q_i = \mathbb{E}[R \mid \mathcal{F}_i] = 1 - u_i(1 - r_i) = (1 - u_i) + u_i r_i. \tag{14}$$

We claim that for each $n \in [N]$ and $i \in [K]$,

$$\mathbb{E}\left[ w_{n,i}^{\mathrm{CAL}} \mid \mathcal{G}_i \right] = u_i = \prod_{j=i+1}^{K} (1 - \rho_j). \tag{15}$$

To see this, define $W_{n,t} := \prod_{j=t}^{K} (1 - \rho_{n,j})$ for $t \in \{i+1, \dots, K\}$, so that $w_{n,i}^{\mathrm{CAL}} = W_{n,i+1}$. We prove by backward induction that

$$\mathbb{E}[W_{n,t} \mid \mathcal{G}_{t-1}] = \prod_{j=t}^{K} (1 - \rho_j), \qquad t = i+1, \dots, K.$$

For the base case $t = K$, conditional on $\mathcal{G}_{K-1}$ the set $\mathcal{S}_K$ is fixed and, for each $m \in \mathcal{S}_K$, $r_{m,K}$ is an i.i.d. Bernoulli($\rho_K$) draw independent of $\mathcal{G}_{K-1}$. Thus $\rho_{n,K}$ is an average of i.i.d. Bernoulli($\rho_K$) variables (with or without leaving out $n$), and therefore $\mathbb{E}[\rho_{n,K} \mid \mathcal{G}_{K-1}] = \rho_K$, i.e. $\mathbb{E}[1 - \rho_{n,K} \mid \mathcal{G}_{K-1}] = 1 - \rho_K$, which is exactly the desired identity at $t = K$.

For the induction step, assume the identity holds at $t + 1$: $\mathbb{E}[W_{n,t+1} \mid \mathcal{G}_t] = \prod_{j=t+1}^{K} (1 - \rho_j)$. Then, by the tower property,

$$\mathbb{E}[W_{n,t} \mid \mathcal{G}_{t-1}] = \mathbb{E}\big[\mathbb{E}[(1 - \rho_{n,t}) W_{n,t+1} \mid \mathcal{G}_t] \big| \mathcal{G}_{t-1}\big]$$

$$= \mathbb{E}\big[(1 - \rho_{n,t})\, \mathbb{E}[W_{n,t+1} \mid \mathcal{G}_t] \big| \mathcal{G}_{t-1}\big]$$

$$= \Big( \prod_{j=t+1}^{K} (1 - \rho_j) \Big) \cdot \mathbb{E}[1 - \rho_{n,t} \mid \mathcal{G}_{t-1}].$$

The same argument as in the base case yields $\mathbb{E}[\rho_{n,t} \mid \mathcal{G}_{t-1}] = \rho_t$ (because the attempt-$t$ outcomes among $\mathcal{S}_t$ are conditionally i.i.d. Bernoulli($\rho_t$) given $\mathcal{G}_{t-1}$, so $\mathbb{E}[1 - \rho_{n,t} \mid \mathcal{G}_{t-1}] = 1 - \rho_t$ and therefore

$$\mathbb{E}[W_{n,t} \mid \mathcal{G}_{t-1}] = \prod_{j=t}^{K} (1 - \rho_j).$$

Setting $t = i + 1$ gives (15).

Using (14), the AL leave-one-out advantage at attempt $i$ can be rewritten as

$$A_{n,i}^{\mathrm{AL}} := Q_{n,i} - \frac{1}{M_i - 1} \sum_{\substack{m \in \mathcal{S}_i \\ m \neq n}} Q_{m,i} = u_i \Big( r_{n,i} - \frac{1}{M_i - 1} \sum_{\substack{m \in \mathcal{S}_i \\ m \neq n}} r_{m,i} \Big) = u_i A_{n,i}^{\mathrm{CAL}},$$

since the constant term $(1 - u_i)$ cancels in the leave-one-out difference. Consequently,

$$Z_i^{\mathrm{AL}} = \frac{1}{M_i} \sum_{n \in \mathcal{S}_i} A_{n,i}^{\mathrm{AL}} g_{n,i} = u_i \cdot \frac{1}{M_i} \sum_{n \in \mathcal{S}_i} A_{n,i}^{\mathrm{CAL}} g_{n,i}. \tag{16}$$

On the other hand, $A_{n,i}^{\mathrm{CAL}} g_{n,i}$ is $\mathcal{G}_i$-measurable, so by (15),

$$\mathbb{E}[Z_i^{\mathrm{CAL}} \mid \mathcal{G}_i] = \mathbb{E}\left[ \frac{1}{M_i} \sum_{n \in \mathcal{S}_i} w_{n,i}^{\mathrm{CAL}} A_{n,i}^{\mathrm{CAL}} g_{n,i} \bigg| \mathcal{G}_i \right]$$

$$= \frac{1}{M_i} \sum_{n \in \mathcal{S}_i} A_{n,i}^{\mathrm{CAL}} g_{n,i} \cdot \mathbb{E}[w_{n,i}^{\mathrm{CAL}} \mid \mathcal{G}_i]$$

$$= u_i \cdot \frac{1}{M_i} \sum_{n \in \mathcal{S}_i} A_{n,i}^{\mathrm{CAL}} g_{n,i} = Z_i^{\mathrm{AL}}, \tag{17}$$

where the last equality uses (16). Taking expectations in (17) yields

$$\mathbb{E}[Z_i^{\text{CAL}}] = \mathbb{E}[Z_i^{\text{AL}}], \qquad \forall i \in [K]. \tag{18}$$

From (12) and (18),

$$\mathbb{E}\left[\widehat{G}_{\text{RLOO}}^{\text{CAL}}(\theta; x, y)\right] = \sum_{i=1}^{K} \mathbb{E}\left[\frac{M_i}{N} Z_i^{\text{CAL}}\right] = \sum_{i=1}^{K} \mathbb{E}\left[\frac{M_i}{N} Z_i^{\text{AL}}\right] = \mathbb{E}\left[\widehat{G}_{\text{RLOO}}^{\text{AL}}(\theta; x, y)\right].$$

Finally, the standard policy-gradient/Doob argument gives $\mathbb{E}[\widehat{G}_{\text{RLOO}}^{\text{AL}}] = \nabla_\theta \mathbb{E}[R] = \nabla_\theta \rho_\theta(x, y)$: indeed,

$$\nabla_\theta \rho_\theta(x, y) = \nabla_\theta \mathbb{E}[R] = \mathbb{E}\left[R \sum_{i=1}^{K} g_i\right] = \sum_{i=1}^{K} \mathbb{E}[Q_i g_i],$$

and the leave-one-out baseline used to form $Z_i^{\text{AL}}$ does not change the expectation because its multiplier has conditional mean zero. This proves (13). □

