# OpenReview forum: "Learning to Correct: Reinforcement Learning for Multi-Attempt Chain-of-Thought"
_ICML.cc/2026/Conference — ICML 2026 regular_

### Official Review · Reviewer_eAGV · 2026-03-05

**Soundness:** 3
**Presentation:** 3
**Significance:** 3
**Originality:** 3
**Overall Recommendation:** 4
**Confidence:** 2

**Summary:**

This paper introduces a framework for optimizing LLMs in multi-attempt reasoning tasks, formalized as the Ver@K problem, where the model has up to K attempts to solve a problem with binary verifier feedback after each try. The authors identify that the trajectory-level (TL) algorithm has high variance, while a naive attempt-level (NAL) reward assignment introduces bias by over-emphasizing early attempts. They propose a calibrated attempt-level (CAL) algorithm, which reweights the gradient contribution of each attempt, yielding unbiased gradients with low variance.

**Compliance With Llm Reviewing Policy:**

Affirmed.

**Key Questions For Authors:**

See above.

**Limitations:**

See above.

**Strengths And Weaknesses:**

**Strengths**

1. The paper theoretically proves that attempt-level optimization has variance less than or equal to that of trajectory-level optimization.
2. It proposes the CAL method to address the practical drawback of naive attempt-level optimization.

**Weaknesses**

1. I find the notation in the paper confusing; a table summarizing the notations used would be helpful.
2. The theoretical guarantees and the design of CAL are developed for a deterministic, binary (pass/fail) verifier. Whether the guarantees of unbiasedness and variance reduction still hold with a noisy or learned verifier (more common in practice), remains an open question.
3. The experiments are conducted with a relatively small K. Scaling the approach to larger K values, which is important for genuinely long CoT, remains to be studied.

---

> ### Author Rebuttal · Authors · 2026-03-31
>
> We thank the reviewer for reviewing our paper and their constructive feedback. In response to their points, we prepared a notation table, extended our analysis to noisy/learned verifier, and added further Ver@4 results for Maze tasks, which can be found in the [following link](https://anonymous.4open.science/r/Learning-to-Correct-Calibrated-Reinforcement-Learning-for-Multi-Attempt-Chain-of-Thought-F6D9/icml_rebuttal.pdf).
> ### Weakness 1
> > I find the notation in the paper confusing; a table summarizing the notations used would be helpful.
>
> **Response:** Thanks for this feedback. We prepared a detailed notation table and added it at the beginning of the appendix. We referenced this table in Section 2.
> ### Weakness 2
> > The theoretical guarantees and the design of CAL are developed for a deterministic, binary (pass/fail) verifier. Whether the guarantees of unbiasedness and variance reduction still hold with a noisy or learned verifier (more common in practice), remains an open question.
>
> **Response.**  Thank you for raising this point. We agree with the reviewer that the application of our work to noisy or learned verifiers is an important question. Under a frozen noisy binary verifier and learned fixed verifier, modeled as a stochastic channel $q_i(\tilde r_i \mid s_{i-1}, y_i),$ which is independent of the policy parameters, an analogue of Theorem 3.1 still holds for the noisy Ver@\(K\) objective $\tilde \rho_\theta(x,y) = \mathbb{E}[\tilde r_{\tilde T}],$ where $\tilde T$ is the stopping time induced by the noisy verifier. The key observation is that our Appendix-C proof is already written in terms of the Doob conditional success probability $Q_i = \mathbb{E}[R \mid \mathcal{F}_i].$ Determinism is only used there to obtain the explicit recursion $Q_i = r_i + (1-r_i) V _{\theta,i}.$ Therefore, if we replace the reward $R$ by the observed verifier reward $\tilde R$, and replace $Q_i$ by $\tilde Q_i = \mathbb{E}[\tilde R \mid \mathcal{F}_i],$ the same argument yields $$\mathbb{E}[\hat G ^{TL} _{\mathrm{RLOO}}] = \mathbb{E}[\hat G^{AL} _{\mathrm{RLOO}}] = \nabla _\theta \tilde \rho _\theta(x,y),$$ and the per-attempt variance-reduction guarantee is unchanged.
>
> For CAL, under the same attempt-index-independence assumption used in Theorem 5.2, the same substitution $r_i \mapsto \tilde r_i$ also preserves unbiasedness for the noisy-verifier objective, because the Appendix-D argument only requires conditional independence of the observed per-attempt outcomes.
>
> Next, we ask whether the noisy verifier induces the same objective as the true verifier, i.e., whether $\tilde\rho_\theta(x,y)=\rho_\theta(x,y)$. In general, it does not, because the verifier controls continuation to the next attempt and therefore changes the effective weighting over attempts. A simple counterexample is $K=2$: suppose the first attempt is correct with probability $p$, and the verifier has type-1 and type-2 error rates $q_1$ and $q_2$. Under the true verifier, the expected number of second attempts is $1-p$, whereas under the noisy verifier it is $p q_2+(1-p) q_1$. Since trajectory-level optimization is unbiased, and it places no additional weighting across attempts, these two verifiers should match $1-p = p q_2+(1-p) q_1$,
> which implies $p = q_2/(q_1+q_2).$ But $p$ changes throughout training, while $q_1$ and $q_2$ are fixed. Hence this equality cannot hold in general over the course of training, so unbiasedness with respect to the true objective is not preserved globally.
>
> We added this extension to Appendix and refer to this extension in Section 2.
>
> ### Weakness 3
>
> > The experiments are conducted with a relatively small K. Scaling the approach to larger K values, which is important for genuinely long CoT, remains to be studied.
>
> **Response:** Thanks for raising this point. We agree that scaling to larger $K$ is important for genuinely long-CoT settings. Our experiments already study both Ver@2 and Ver@4 settings, and in the rebuttal we further added Ver@4 results on the $7\times 7$ and $9\times 9$ Maze tasks, please see “Additional Ver@4 7x7 Maze Results” and “Additional Ver@4 9x9 Maze Results” in the attached rebuttal note. In both settings, CAL-GRPO attains the best Ver@4 among the GRPO baselines, which suggests that its benefit persists when moving from $K=2$ to $K=4$.
>
> We also agree that studying substantially larger $K$ is an important future direction, but it quickly becomes computationally expensive for two reasons. (1) Larger $K$ values require longer rollouts as the context length is growing linearly with $K$. (2) Larger $K$ values demand larger group sizes to estimate the calibration weights reliably at each attempt. Please see ablation studies on group sizes in Figure 2 in the link.
>
> As a future direction, context management techniques could be developed for the first problem such as summarizing the earlier attempts. For the second problem, estimation of the calibration weights with smaller group size can be further analyzed.

---

> > ### Author Rebuttal · Reviewer_eAGV · 2026-04-03
> >
> > My concerns have been adequately addressed. I wiil keep my score.

---

> > > ### Author Response · Authors · 2026-04-06
> > >
> > > Thank you again for your thoughtful feedback and for marking the concerns as fully resolved. If you feel that the revisions and additional results have adequately addressed your concerns, we would be grateful if you would consider updating your overall assessment. Of course, we understand that this is entirely at your discretion.

---

### Official Review · Reviewer_XX86 · 2026-03-11

**Soundness:** 3
**Presentation:** 2
**Significance:** 3
**Originality:** 3
**Overall Recommendation:** 4
**Confidence:** 5

**Summary:**

This work tries to apply reinforcement learning to language models for complex multi-attempt reasoning tasks. The authors reveal a central problem of credit assignment when models receive sparse verifier feedback after multiple retries. To address this, they propose a Calibrated Attempt-Level (CAL) algorithm that dynamically weights the reward of each attempt based on future success probabilities. This approach theoretically guarantees unbiased, low-variance policy gradients and empirically demonstrates superior Verification@K performance across math, maze navigation, and Markov Chain benchmarks.

**Compliance With Llm Reviewing Policy:**

Affirmed.

**Final Justification:**

My concerns have been addressed. I decide to keep this possitive rating.

**Key Questions For Authors:**

1.In Line270, is Markov Chain Model your own define benchmark? If yes, please pur more details and cases. If it is existing benchmark, please include the citation. Also, the meaning of ‘Model’ in Line 270 is different from that in Line 311, right?

2.‘For the Markov chain task, we initialize the policy from an SFT checkpoint.’ The detailsare missin.

**Limitations:**

yes

**Strengths And Weaknesses:**

**Strength:**

1. By addressing the credit assignment problem in multi-retry settings, this paper demonstrates significant practical engineering value for multi-turn RL training.

2. The CAL algorithm does not require modifying the architecture of large language models. It simply alters the weighting rules of the Advantage function during the loss calculation phase in RL (e.g., GRPO), resulting in low implementation costs.

**Weakness:**

1. Concern about Over-packaging. 1) This paper frames a Verification@K problem, which is exaclty the popular multi-turn RL. 2) The TL and NAL are claimed as variants of CAL, which actually are standard credit assignment methods in current multi-turn GRPO works. I think you should mention these related works in the introduction/relatedwork and baseline setting, rather than framing it as a newly defined problem. For example, some works[1,2,3,4,5,etc] have explored multi-turn GRPO training and reward design/reweighting. Or rather, your approach is fundamentally different from them.

2. The contribution is not clear enough. I think your Calibration Design could be attractive is you put more effort on Eq.10 and related ablation and explanation. However, there two many Formulas and expressions that distract the reader so much that they don't grasp your core contribution. Meanwhile, Sec.4 is also not clear. It is hard to extarct your core conclusion, analysis and ablation from the plain expression.

[1] MURPHY: Multi-Turn GRPO for Self Correcting Code Generation

[2] Turn-Level Trajectory Optimization for Robust Multi-Turn LLM Reasoning

[3] Empowering Multi-Turn Tool-Integrated Reasoning with Group Turn Policy Optimization

[4] Reinforcing Multi-Turn Reasoning in LLM Agents via Turn-Level Credit Assignment

[5] Reinforcing Multi-Turn Reasoning in LLM Agents via Turn-Level Reward Design

**Issue:**

Impact Statement is missing.

The overall feeling of this paper is difficult to grasp core contribution and conclusion. I will consider raising score to positive if the weakness is addressed.

---

> ### Author Rebuttal · Authors · 2026-03-31
>
> We thank the reviewer for reviewing our paper and their constructive feedback. In response to reviewer's comments, we improved our presentation of CAL and Section 4 by dividing into multiple parts. Furthermore, we improved our related works section by incorporating multi-turn RL works and added the impact statement, both of which can be found in the [following link](https://anonymous.4open.science/r/Learning-to-Correct-Calibrated-Reinforcement-Learning-for-Multi-Attempt-Chain-of-Thought-F6D9/icml_rebuttal.pdf).
>
> ### Weakness 1
>
> > The paper seems over-packaged relative to prior multi-turn RL work. Please better position Ver@K, TL, and NAL with respect to existing turn-level GRPO methods and clarify what is genuinely new.
>
> **Response:** Thank you for raising this point. We agree that turn-/attempt-level reward design has also been explored in prior multi-turn RL work, and we do not intend to claim those ingredients as new. We revised our framing accordingly to make it clearer that TL and NAL are standard baselines. Our main contribution is to show that naive attempt-level reward assignment is biased for the Ver@K objective, and to correct this with CAL-GRPO, which calibrates attempt weights to better optimize the Ver@K objective. We also expanded the related-work discussion with a new “Multi-turn RL and fine-grained credit assignment” paragraph and added two of the additional works you mentioned; this appears in the attached rebuttal PDF.
>
> ### Weakness 2
>
> > Concern about clarity/presentation. The core contribution, especially Eq. 10 and the calibration design, is not clear enough. There are too many formulas, and Section 4 makes it hard to extract the main conclusions and ablations.
>
> **Response:** Thank you for this helpful comment. We agree that the main message would benefit from a clearer presentation, and we revised the paper accordingly. We split Section 4 into **Experimental Setup** and **Results and Analysis**, and organized the discussion so that each paragraph begins with a bold "takeaway message". The results subsection now begins by explicitly stating the three questions we study whether attempt-level credit assignment improves over trajectory-level training, whether calibration beyond naive attempt-level normalization is necessary, and how calibration affects the accuracy–compute tradeoff. We clarified Eq. (10) by stating directly that CAL reweights attempt $t$ by the estimated probability that all later attempts fail, i.e., its marginal contribution to final Ver@K.
>
> ### Issue 1
>
> > Impact Statement is missing.
>
> **Response:** We appreciate the reviewer for highlighting this. We have added the Impact Statement to the rebuttal PDF.
>
> ### Q1
>
> > 1.In Line270, is Markov Chain Model your own define benchmark? If yes, please pur more details and cases. If it is existing benchmark, please include the citation. Also, the meaning of ‘Model’ in Line 270 is different from that in Line 311, right?
>
> **Response:** Thanks for flagging this. Yes, the Markov chain task is our own synthetic benchmark, not an existing benchmark. We introduced it as a controlled evaluation setting for multi-attempt reasoning, where the task structure and verifier can be specified precisely. We will make this more explicit in the main text and point readers to Appendix A, where the benchmark is described in more detail. Concretely, it is a structured Markov-chain planning task with ordered hubs, sparse boundary transitions, an absorbing terminal state, and two settings: a standard no-trap version and an optional trap variant. Success requires producing a valid shortest path, and under Ver@K, failed attempts reset to the same start state. Finally, we agree that the term “model” is ambiguous here, in Line 270 it refers to the benchmark/task, whereas in Line 311 it refers to the policy model. We will revise Line 270 to “Markov chain task” to avoid confusion.
>
> We also agree that adding further details would make the benchmark clearer, and in the revision we will expand the description and add visualizations to improve clarity.
>
> ### Q2
>
> > 2.‘For the Markov chain task, we initialize the policy from an SFT checkpoint.’ The detailsare missin.
>
> **Response:** Thank you for pointing this out. We already give brief details about the SFT training data immediately after that sentence in the main text, but we agree that this part should be more explicit. In the revision, we will move key rollout and training details from Appendix A to the main text and expand the appendix accordingly. Concretely, for the Markov chain task, we first train an SFT checkpoint on synthetic optimal trajectories. We sample start states (and the trap choice in the trap setting), compute the corresponding shortest valid trajectory to the absorbing terminal state, and use these trajectories as supervised targets. We then fine-tune this checkpoint with the same Ver@K GRPO procedure for all three estimators.

---

> > ### Author Rebuttal · Reviewer_XX86 · 2026-04-03
> >
> > My main concerns are addresses. I will maintain this possitive score.

---

> > > ### Author Response · Authors · 2026-04-06
> > >
> > > Thank you again for your thoughtful feedback and for marking the concerns as fully resolved. We really appreciate that you provided clear criteria to increase your score. If you feel that the revisions and additional results have adequately addressed your concerns, we would be grateful if you would consider updating your overall assessment. Of course, we understand that this is entirely at your discretion.

---

### Official Review · Reviewer_g8D9 · 2026-03-13

**Soundness:** 3
**Presentation:** 3
**Significance:** 2
**Originality:** 4
**Overall Recommendation:** 4
**Confidence:** 4

**Summary:**

The authors present CAL-GRPO which is a useful gradient estimator in an interesting setting where the model gets at most K attempts to answer a prompt correctly and the reward is 0/1. The estimator is unbiased under some assumptions and performs well empirically on some synthetic and real math reasoning benchmarks.

**Compliance With Llm Reviewing Policy:**

Affirmed.

**Final Justification:**

My concerns have been addressed.

**Key Questions For Authors:**

I generally like the paper and only have 1 request from the authors, post which I would be willing to increase my score.

1. Could the authors redo the experiment without the std. deviation normalization in GRPO and report the results? I feel the theory has all been made for the case where the std. deviation is absent but the experiments have the std. deviation term. I think this weakens the paper a little.
2. "GRPO and RLOO can be equivalent under appropriate assumptions, see Thrampoulidis et al. (2025)". I have major reservations against this statement because the surrogate reward, for which GRPO and RLOO are equivalent, keeps changing during training. I would request the authors to remove this and do the experiment in the other case (no std. deviation).
3. This is just a general question (feel free to skip it) : is it possible to study the effects of bias of the estimator? Could it lead to pathological behavior?

**Limitations:**

yes

**Strengths And Weaknesses:**

1. The estimator is easy to implement in practice and has zero overhead.
2. The theory is elegant, although the estimator is biased in the general case, the authors found some assumptions where the estimator is unbiased and works well empirically.

---

> ### Author Rebuttal · Authors · 2026-03-31
>
> We thank the reviewer for reviewing our paper and their constructive feedback. In response to reviewer's comments, we added ablation studies where we disable the standard deviation normalization for the advantage calculation. The figures can be found in the [following link](https://anonymous.4open.science/r/Learning-to-Correct-Calibrated-Reinforcement-Learning-for-Multi-Attempt-Chain-of-Thought-F6D9/icml_rebuttal.pdf).
>
> ### Q1
>
> > Could the authors redo the experiment without the std. deviation normalization in GRPO and report the results? I feel the theory has all been made for the case where the std. deviation is absent but the experiments have the std. deviation term. I think this weakens the paper a little.
>
> **Response:** Thanks for raising this question. We agree with the reviewer that there is a slight difference between the theoretical derivation of the calibrated attempt-level optimization and CAL-GRPO in the sense that the theoretical derivation uses only mean normalization with leave-one-out, whereas the CAL-GRPO uses both mean and std normalization. We provide an ablation study in Figure 3, where we apply only mean normalization to the TL-GRPO, NAL-GRPO, and CAL-GRPO on no-trap Markov Chain tasks and 5x5 and 9x9 Maze tasks. Our observation from Figure 3 is that std normalization improves all baselines and it does not change the performance ordering of the baselines. We incorporate this ablation study in Appendix. In fact, we had tried both normalization variants before submitting this paper and we had decided to continue with the standard deviation normalization, as mean normalization is suboptimal compared to mean+std normalization.
>
> ### Q2
>
> > "GRPO and RLOO can be equivalent under appropriate assumptions, see Thrampoulidis et al. (2025)". I have major reservations against this statement because the surrogate reward, for which GRPO and RLOO are equivalent, keeps changing during training. I would request the authors to remove this and do the experiment in the other case (no std. deviation).
>
> **Response:** Thanks for the question. We agree with the reviewer that there is a slight divergence between these two algorithms. In response to reviewer's point, we decided to remove this footnote. Instead, we added the following footnote, which makes the paper more coherent: "Theoretical derivations of the algorithms apply only mean normalization through leave-one-out, whereas GRPO variants of these algorithms apply both mean and std normalizations. We observe that mean+std normalization performs better than plain mean normalization and provide the experimental results with the mean+std deviation. We provide some ablation studies in Appendix B where we only apply mean normalization."
>
> ### Q3
>
> > This is just a general question (feel free to skip it) : is it possible to study the effects of bias of the estimator? Could it lead to pathological behavior?
>
> **Response:** Thanks for the insightful question. We believe the Naive Attempt-Level (NAL) algorithm provides a strong example of estimator bias in practice and how it leads to a pathological behavior. Under the independent attempt-index assumption (Assumption 5.1), Theorem 5.2 shows that CAL is unbiased. Building on this, Corollary 5.3 shows that NAL effectively optimizes a different objective, one that places greater weight on earlier attempts than on later ones. In other words, NAL is biased toward early attempts. Indeed, we observe this behavior in our empirical results. In Figures 2b and 4b, NAL performs better than CAL on the first attempt, even though its performance on the Ver@2 objective is worse. This suggests that NAL prioritizes improving earlier attempts in a way that is misaligned with the Ver@2 objective. Furthermore, in response to Reviewer TKcF, we provide an empirical comparison of bias and variance between different algorithms in Figure 1 in [this link](https://anonymous.4open.science/r/Learning-to-Correct-Calibrated-Reinforcement-Learning-for-Multi-Attempt-Chain-of-Thought-F6D9).

---

> > ### Author Rebuttal · Reviewer_g8D9 · 2026-04-04
> >
> > My concerns have been addressed.

---

> > > ### Author Response · Authors · 2026-04-06
> > >
> > > Thank you again for your thoughtful feedback and reviewing our paper.

---

### Official Review · Reviewer_TKcF · 2026-03-18

**Soundness:** 4
**Presentation:** 4
**Significance:** 3
**Originality:** 3
**Overall Recommendation:** 5
**Confidence:** 4

**Summary:**

TLDR: The paper recasts multi-attempt LLM verification as a random-horizon problem where the probability of failure on the verifier plays the role of the discount factor, then Rao-Blackwellizes over attempts for variance reduction. The theory is quite clean, but the experiments don't directly measure the variance reduction and could use group size ablations.

**Compliance With Llm Reviewing Policy:**

Affirmed.

**Key Questions For Authors:**

My key question is really whether this helps in practice. You have learning curves and that's great, but they don't quite isolate the mechanism. Help me make sense of your results under this lens: that of variance reduction, which is ultimately a proxy for credit assignment. I'd like you to think more carefully about what the right metrics would be to properly measure the phenomenon.

**Limitations:**

- Be more upfront about the lack of hard guarantees on variance reduction unless you have special structure (ie. independence) or happen to land on a covariance term which is favorable.

- Better connect to the RL (pre-llm times) literature

**Strengths And Weaknesses:**

I enjoyed reading this paper. It's clean, well-written, and properly grounded in RL theory, which I appreciate.

A few points:
The attempt vs. trajectory distinction is a bit confusing as stated. It would help to anchor these notions into concrete examples early on in the text: eg. code generation, math problem solving, whatever fits, so the reader can build intuition before the formalism kicks in.

Now, the main idea. Here's really what this is about imo. In discounted MDPs, you can interpret the infinite-horizon discounted objective as a finite-horizon problem with a random horizon, where the horizon follows a geometric distribution: continuation probability $\gamma$, stopping probability 1 − $\gamma$. That's essentially what's happening here, except that continuation or stopping is governed by the interaction of the model with the verifier. While the verifier in practice is deterministic and only provide binary feedbac, its interaction with the model also controls whether the process continues: the combination of both is essentially the "environment" in RL. The main idea is then one of marginalizing out the trajectory structure within each attempt, which is the essence of conditional Monte Carlo or Rao-Blackwellization for variance reduction which we have seen in a many papers in RL (eg. all the "expected this", "expected that" variants in RL, hindsight credit assignment, marginalized importance sampling etc.). This is what Theorem C.1 in the appendix formalizes.
I'll note that the authors have been careful not to fall into a common trap: the theorem states "for each attempt index i," and that's the right thing to say. It would be worth highlighting why this per-attempt qualification is necessary. The variance of a sum of random variables is not just the sum of the variances unless the covariance terms vanish, which is rarely the case without very specific independence assumptions. Does this mean the trick doesn't help in practice? Absolutely not. If you can marginalize like that, it almost always pays off empirically. But it's good to be transparent about the gap between the per-attempt guarantee and the full objective.

From there, the authors derive policy gradient estimators based on this, with the RLOO variants following naturally.
One more conceptual connection worth making: as opposed to the discount factor $\gamma$ in standard MDPs, what plays the role of a "discount" here is the probability of failure (i.e., of continuing to the next attempt). This probability is not known ahead of time, so the authors estimate it from empirical frequencies within a group. This whole idea of learning a continuation probability is reminiscent of the "cumulant" concept in the General Value Function (GVF) framework. The same kind of multiplicative accumulation of continuation probabilities (essentially the exponentiation to the power $t$ that you get with fixed discounting) appears in that line of work as well. More for inspiration than necessity, but it would be good to cite and connect to this classic work (Sutton et al., 2011).

Regarding the experiments: unfortunately, they don't quite isolate the effect of this trick as a variance reduction method. I wonder if this could be done more explicitly, for instance, by measuring the empirical variance of the gradient estimator and checking whether it does go down compared to the baseline estimator. It would also be nice to have an ablation that goes beyond showing a curve as a function of training steps. Specifically, a plot showing performance as a function of the group size would be informative, since the quality of the continuation probability estimate depends directly on it.

---

> ### Author Rebuttal · Authors · 2026-03-31
>
> We thank the reviewer for reviewing our paper and their constructive feedback. We are glad that the reviewer enjoyed reading the paper. In response to reviewer's point, we empirically measure the bias and variance of the gradient estimators across different baselines and add ablation studies on rollout group size. The figures are available in the linked [supplementary note](https://anonymous.4open.science/r/Learning-to-Correct-Calibrated-Reinforcement-Learning-for-Multi-Attempt-Chain-of-Thought-F6D9/icml_rebuttal.pdf).
>
> ### Q1
>
> > The attempt vs. trajectory distinction is a bit confusing as stated.
>
> **Response:** Thanks for the feedback. We provided an example with a math question in the introduction where we now explain the attempt-level approach.
>
> ### Q2
>
> > I wonder if this could be done more explicitly, for instance, by measuring the empirical variance of the gradient estimator and checking whether it does go down compared to the baseline estimator.
>
> **Response:** Great question! This question made us think about a way to measure the actual bias and variance of the gradient estimators, which, we believe, substantially improves our paper. Due to model size, we focus on the trap Markov-chain setting. We sample some models from the training of CAL-GRPO methods from steps 500 until 3000. (The same model across different baselines.) Theorem 3.1. proves that trajectory-level optimization is unbiased, and TL-GRPO directly corresponds to trajectory-level optimization. Using the law of large numbers, we estimate the true gradients of each model by sampling $4800\times20\times16$ trajectories and taking the average of TL-GRPO gradients. Then, based on the true gradient estimate, we calculate the overall bias, overall variance of the trajectories in addition to the individual variances of attempts 1 and 2 for different baselines. We provide these results in Figure 1 in the link above. Figure 1(a) demonstrates the overall bias order as TL-GRPO \< CAL-GRPO \< NAL-GRPO, which is consistent with Theorem 5.2 and Corollary 5.3. Figure 1(b) demonstrates the overall variance order as CAL-GRPO \< NAL-GRPO ~ TL-GRPO in parallel to Theorem 3.1. Importantly, each attempt tries to solve the same task, which means that the cross-covariance between each attempt's gradient is pretty high. This transfers the variance reduction property from each attempt to the overall trajectory as the reviewer pointed out (please see line 211-213 in the paper). Figures 1(c) illustrates the variance reduction for the first attempt in the estimation of the gradient in parallel to Theorem 3.1. It implies that the estimation of attempt-level optimization with CAL-GRPO works well in the practical settings. Figures 1(d) provides the equality condition in Theorem 3.1. (There is a typo in Theorem 3.1. The variance reduction should hold for $i \in [K-1]$. Please note that $i=K$ reduces to $K=1$, which is the equality case.) Indeed, until step 2000, the variances of TL-GRPO and CAL-GRPO are equivalent. This implies that the estimation of attempt-level optimization with CAL-GRPO works pretty well. However, after step 2000, CAL-GRPO estimate becomes less accurate as the second attempt becomes highly dependent on the first attempt due to trap, which violate the independent attempt-index assumption (Assumption 5.1). Please observe that step 2000 corresponds to the end of the plateau in Figures 3(b) in the main paper, where the second attempt's success is highly dependent on the first attempt.
>
> ### Q3
>
> > Specifically, a plot showing performance as a function of the group size would be informative, since the quality of the continuation probability estimate depends directly on it.
>
> **Response:** Thanks for the question. We provide a group-size ablation for the no-trap Markov-chain task and the 5×5 and 7×7 Maze tasks in Figure 2. As the reviewer suggests, the quality of CAL-GRPO’s calibration depends on the rollout group size. As expected, decreasing the group size degrades CAL-GRPO’s performance. However, smaller groups also degrade the other baselines, and CAL-GRPO still performs best overall. This is because all three methods are group-relative estimators, so smaller groups reduce gradient-estimation quality more broadly. Overall, the ablation suggests that CAL-GRPO requires a group size comparable to those used by the standard baselines.
>
> ### Q4
>
> > Better connect to the RL (pre-llm times) literature
>
> **Response:** We thank the reviewer for this insightful connection. We added the following after eq (11): "(11) can be interpreted through the lens of General Value Functions (GVFs) (Sutton et al., 2011), where the continuation probability plays a role analogous to a state-dependent discount factor, and the attempt-level signal corresponds to a cumulant. In this view, our method can be seen as learning a GVF with an empirically estimated continuation function, which induces the same multiplicative structure over time."

---

### Decision · Program_Chairs · 2026-04-30

**Decision:**

Accept (regular)

**Comment:**

The paper presents a principled RL formulation for multi-attempt reasoning and proposes CAL, a calibrated attempt-level credit assignment method for optimizing Ver@K. Reviewers agreed that the work is technically solid, with clean theory, practical relevance, and low implementation overhead, and they found the empirical results broadly supportive. The main weaknesses were clarity and positioning. The authors addressed these concerns well in the rebuttal by adding bias/variance analyses, group-size and normalization ablations, clearer discussion of related work, improved explanation of the core contribution, and extensions to noisy verifiers and larger K. Multiple reviewers explicitly stated that their concerns were fully resolved and all final scores remained positive. Overall, this is a solid submission that merits acceptance.